# Increasing plant diversity with border crops reduces insecticide use and increases crop yield in urban agriculture

**Nian-Feng Wan[1,2†], You-Ming Cai[1†], Yan-Jun Shen[3], Xiang-Yun Ji[1], Xiang-Wen Wu[4], Xiang-Rong Zheng[2], Wei Cheng[4], Jun Li[5], Yao-Pei Jiang[4], Xin Chen[6], Jacob Weiner[7], Jie-Xian Jiang[1]\*, Ming Nie[2], Rui-Ting Ju[2], Tao Yuan[1], Jian-Jun Tang[6], Wei-Dong Tian[2], Hao Zhang[1], Bo Li[2]\***

[1]Eco-environmental Protection Research Institute, Shanghai Academy of Agricultural Sciences, Shanghai Key Laboratory of Protected Horticultural Technology, Shanghai Engineering Research Centre of Low-carbon Agriculture, Shanghai, China; [2]Ministry of Education Key Laboratory for Biodiversity Science and Ecological Engineering, Shanghai Chongming Dongtan Wetland Ecosystem Research Station, Institute of Biostatistics, Shanghai Institute of Eco-Chongming, (SIEC), Fudan University, Shanghai, China; [3]Chongming Agricultural Technology Extension and Service Center, Shanghai, China; [4]Shanghai Agricultural Technology Extension and Service Center, Shanghai, China; [5]Climate Center of Shanghai, Shanghai, China; [6]College of Life Sciences, Zhejiang University, Hangzhou, China; [7]Department of Plant and Environmental Sciences, University of Copenhagen, Frederiksberg, Denmark

**\*For correspondence:**
jiangjiexian@163.com (J-XJ);
bool@fudan.edu.cn (BL)

[†]These authors contributed equally to this work

**Competing interests:** The authors declare that no competing interests exist.

**Abstract** Urban agriculture is making an increasing contribution to food security in large cities around the world. The potential contribution of biodiversity to ecological intensification in urban agricultural systems has not been investigated. We present monitoring data collected from rice fields in 34 community farms in mega-urban Shanghai, China, from 2001 to 2015, and show that the presence of a border crop of soybeans and neighboring crops (maize, eggplant and Chinese cabbage), both without weed control, increased invertebrate predator abundance, decreased the abundance of pests and dependence on insecticides, and increased grain yield and economic profits. Two 2 year randomized experiments with the low and high diversity practices in the same locations confirmed these results. Our study shows that diversifying farming practices can make an important contribution to ecological intensification and the sustainable use of associated ecosystem services in an urban ecosystem.
DOI: https://doi.org/10.7554/eLife.35103.001

## Introduction

Over the last century, global biodiversity loss and species extinction have occurred at an unprecedented rate (*Barlow et al., 2016*), and agricultural intensification has been one of the major drivers (*Tscharntke et al., 2005*; *Gonthier et al., 2014*). One of the features of agricultural intensification is the striking change in land use, in which complex natural ecosystems have been converted to monocultural crop production ecosystems (*Tscharntke et al., 2005*). This agricultural intensification has multiple consequences for ecosystems, including a decline in natural biocontrol services (*Symondson et al., 2002*), disruption of crop pollination (*Kremen et al., 2012*; *Kovács-Hostyánszki et al., 2017*) and extensive damage to naturally-occurring species and the environment

from the heavy use of agrochemicals (*Tscharntke et al., 2005*; *Gill et al., 2012*; *Stehle and Schulz, 2015*).

Ecological intensification is a new paradigm for agriculture (*Gaba et al., 2014*; *Simons and Weisser, 2017*), based on intensifying ecological processes in cultivated areas (*Caron et al., 2014*; *Bowles et al., 2017*) to replace synthetic, non-renewable and often toxic anthropogenic inputs with biologically- and environmentally-friendly ecosystem services (*Tscharntke et al., 2005*; *Geertsema et al., 2016*). The goals are to achieve more sustainable crop production (*Bommarco et al., 2013*), increase food security, and improve the quality of agricultural products (*Martin-Guay et al., 2018*).

Biologically-diversified farming can contribute to ecological intensification of agriculture by providing multiple ecosystem services, such as the promotion of biocontrol services (e.g., a decrease in pest abundance and an increase in the abundance of natural enemies of pests) (*Harvey et al., 2014*; *Gurr et al., 2016*), a reduction in negative environmental impacts (*Tscharntke et al., 2005*; *Tittonell, 2014*; *Zhao et al., 2016*; *Barot et al., 2017*), and an increase in crop yields (*Cassman, 1999*; *Tittonell and Giller, 2013*; *Wittwer et al., 2017*). Most previous studies on biodiversity and ecological intensification have focused on smallholder farms (*Tittonell and Giller, 2013*; *Zimmerer, 2013*; *Zimmerer et al., 2015*), while urban agriculture has received little attention.

Starting in the 1950s, urban agriculture, defined as the production of crop and livestock goods within cities and towns, has played an increasing role in ensuring food security for growing urban populations (*Zezza and Tasciotti, 2010*). Urban agricultural systems have many forms, such as community farms, allotment gardens, rooftop gardens, and edible landscaping (*Lin et al., 2015*). Confronted with population growth and a shortage of land and natural resources in urban ecosystems, urban agriculture has depended heavily on intensive use of pesticides and fertilizers (*Altieri et al., 1999*). It has been hypothesized that increased biodiversity in urban agricultural systems has the potential to provide important ecosystem services such as pollination, pest control, and climate resilience, thus reducing the need for chemical inputs (*Lin et al., 2015*). Replicated, multi-site and long-term studies evaluating the effects of plant-diversified farming on ecosystem services have been lacking.

To reduce urban poverty, enhance food security, improve environmental management and stimulate participatory city governance, the Resource Centre on Urban Agriculture and Food Security (RUAF Foundation) has established pilot demonstration cities for modern urban agriculture worldwide. One of these cities is Shanghai, China. Shanghai has a 30 year history as a pioneer in urban agriculture, which has played an important role in the urban economy, social stability and public security (*Kiminami et al., 2006*). This mega-urban municipality includes nine suburbs with agriculture, covering an estimated area of 93,333 hectares of rice, accounting for 14.7% of the total land area in Shanghai. Although agriculture contributes only 0.33% of the city's Gross Domestic Product and increases in productivity have been slower than in other economic sectors, the city authorities have been paying considerable attention to agricultural production to ensure a stable food supply for the huge urban population. In addition, the annual profit from agricultural sightseeing and tourism in Shanghai is 2.4 billion RMB, further increasing the value of the city's agriculture. Community farms account for 80% of the total agricultural land in Shanghai. Despite strict laws and regulations, these farms have been experiencing increased intensification and mechanization, resulting in biodiversity loss, food safety problems and land degradation.

Although there is evidence that diversifying farming promotes ecological intensification in smallholder agricultural systems (*Gurr et al., 2016*; *Rolando et al., 2017*), the ecological and agronomic consequences of increased crop diversity have not been studied in depth in urban agricultural environments. The effects of biodiversity on ecosystem services could be different in an urban setting, because urban farms are not surrounded by other farms or semi-natural vegetation, which can provide organisms that deliver ecosystem services. We hypothesize that diversifying farming in an urban setting will host more insect predators, reducing the densities of herbivorous insects, and allowing reduced input of insecticides without considerable yield loss.

To test our hypothesis, we analyzed monitoring data collected from 34 community farms in nine districts in suburbs of Shanghai, and we conducted two controlled experiments for further verification. In the early 1990s, Shanghai Agricultural Technology Extension and Service Center (SATESC) established 28 community farms in 8 districts of Shanghai to monitor pest occurrence in rice fields. With the help of Chongming Agricultural Technology Extension and Service Center (CATESC),

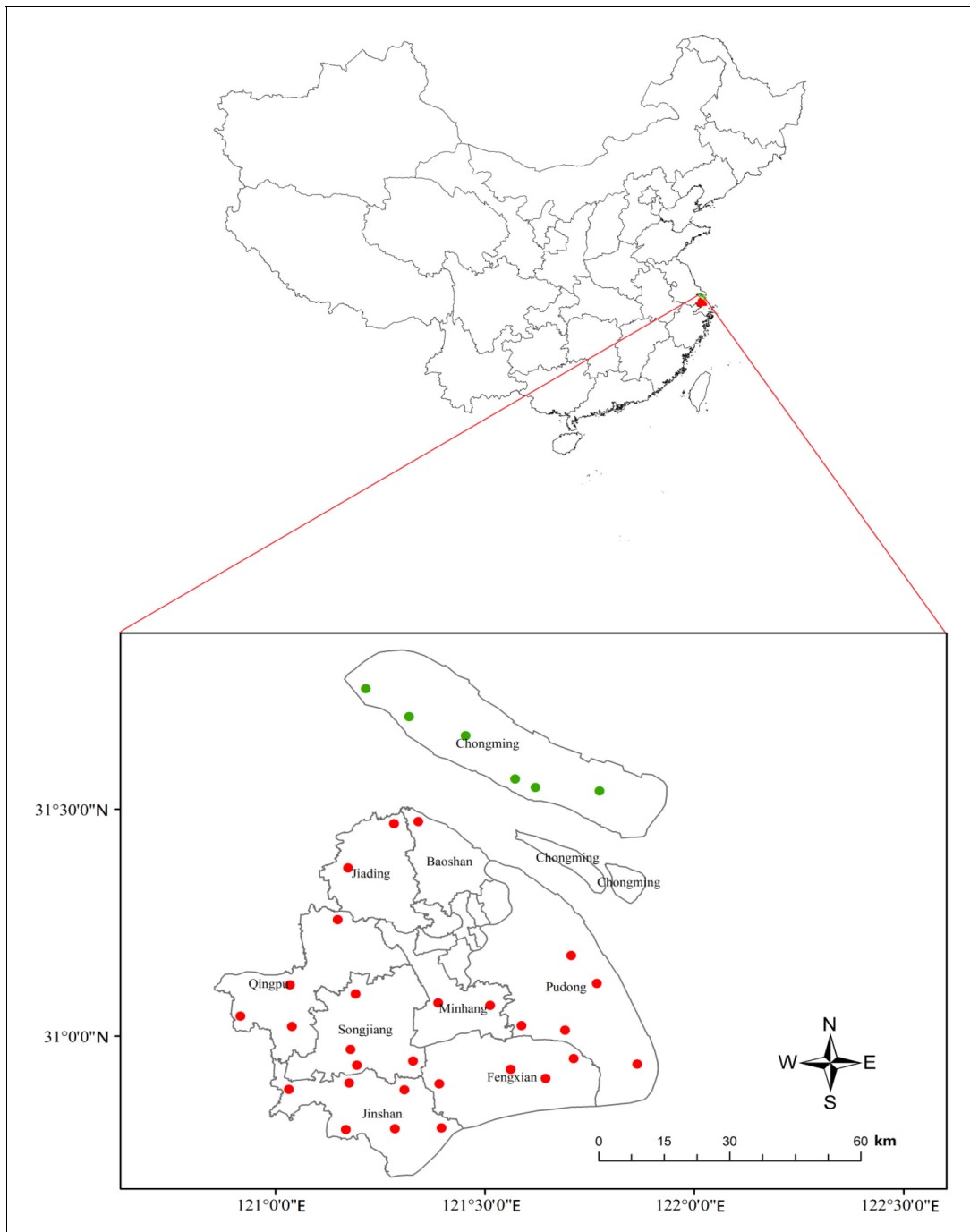

**Figure 1.** Twenty-eight mono-rice (red dots) and six plant-diversified community farms (green dots) monitored in Shanghai, China.
DOI: https://doi.org/10.7554/eLife.35103.002

The following source data is available for figure 1:

**Source data 1.** Site, rice pest and predator populations, insecticide use, and yield data for comparison of plant-diversified farms (treatment) and mono-rice farms (control) in Shanghai, China.
DOI: https://doi.org/10.7554/eLife.35103.003

**Source data 2.** Chemical insecticides applied to control the main insect pests on plant-diversified farms and mono-rice farms in Shanghai, China from 2001 to 2015.
DOI: https://doi.org/10.7554/eLife.35103.004

another six community farms in Chongming district were established for monitoring pest occurrence in rice fields (*Figure 1*; *Figure 1—source data 1*). Through our own investigation, we found that the six community farms in Chongming district all included diversified plants, and the 28 community farms in the other 8 districts of Shanghai consisted of rice monocultures. The farmers of the six community farms in Chongming district planted soybeans as a border crop around the periphery of each rice field and common vegetable crops (maize, eggplant, Chinese cabbage, etc.) around the whole rice-growing area.

The abundances of the three main rice pests, which have resulted in major crop losses since the late 1990 s, were monitored. The abundance of pink rice borer (*Sesamia inferens*) and rice brown planthopper (*Nilaparvata lugens*) in rice fields were evaluated with trapping lamps, and abundance of rice leaf roller (*Cnaphalocrocis medinalis*) was estimated with field surveys. We also estimated the densities of the main predators of these pests in the rice fields and in the border and neighboring crops of the diversified farms, monitored pesticide use and grain yield, and conducted a continuous economic cost–benefit analysis on both farm types. Complete monitoring data for rice pests, pesticide use and grain yield on these farms started in 2001, and monitoring of predators began at the beginning of 2007. Since the diverse and mono-rice farms were located in different areas of the city, group members from the Shanghai Academy of Agricultural Sciences supplemented the monitoring data with two controlled experiments, in which both treatments were performed at two locations over two years in a complete random design.

## Results

### Occurrence of rice pests

For all three pest groups, there were no significant interactions between farm type and year (pink rice borer: LR [Likelihood Ratio]=18.617, p=0.338; rice plant-hopper: LR = 12.772, p=0.684; leaf roller: LR = 7.600, p=0.952). Therefore, we analyzed the effects of plant-diversified versus mono-rice farms with main-effects models with only farm type and year as predictors. The main-effects models showed that the abundances of pink rice borer, rice plant-hopper and leaf roller were significantly lower on plant-diversified farms (pink rice borer: LR = 13.864, p=0.002; rice plant-hopper: LR = 10.361, p=0.004; leaf roller: LR = 12.827, p=0.001). The percent decrease in mean pest abundance on the plant-diversified farms varied from year to year over the 15 year study period (pink rice borer: mean percent decrease = 28.8, SD [standard deviation]=9.9, range = 16.3–47.0, IQR [interquartile range]=20.8–34.5; rice brown plant-hopper: mean percent decrease = 32.3, SD = 17.2, range = 16.8–85.5, IQR = 24.2–30.4; rice leaf rollers: mean percent decrease = 20.0, SD = 6.7, range = 9.0–33.8, IQR = 15.6–24.0) (*Figure 2*; *Figure 2—source datas 1–3*).

### Abundance of predators

For predator abundance (ladybird beetles, lacewings and spiders), there was no significant interaction between farm type and year (LR = 6.150, p=0.740). The main-effects model (with farm type and year as predictors) showed that plant-diversified farms had significantly higher predator abundance than mono-rice farms throughout 2007–2015 (LR = 37.002, p=0.001). From 2007 to 2015, annual average abundance of pests' predators observed during all the four stages of rice development was 22.6% (±3.0% [SE]; range 8.3–32.0%) higher on plant-diversified than on mono-rice farms (*Figure 3*; *Figure 3—source data 1*).

From 2007 to 2014, abundances of these predators in the soybean border crop ranged from 69.4 (±11.6) to 127.4 (±19.9) individuals per 100 plants, and in the neighboring crops it ranged from 121.0 (±27.6) to 153.6 (±26.6) individuals per 100 plants in maize, 101.3 (±21.3) to 117.3 (±22.5) individuals per 100 plants in eggplant, and 39.4 (±6.2) to 46.0 (±5.2) individuals per 100 plants in Chinese cabbage (*Figure 4A*). Annual average abundances of predators in soybean, maize, eggplant and Chinese cabbage were 92.4 (±7.1), 141.9 (±5.7), 107.3 (±2.9) and 43.1 (±1.3) individuals per 100 plants, respectively (*Figure 4B*).

### Insecticide use

There were significant interactions between farm type and year for the number and amount of insecticide sprays (number: LR = 56.395, p=0.001, amount of commercial insecticide: LR = 96.67,

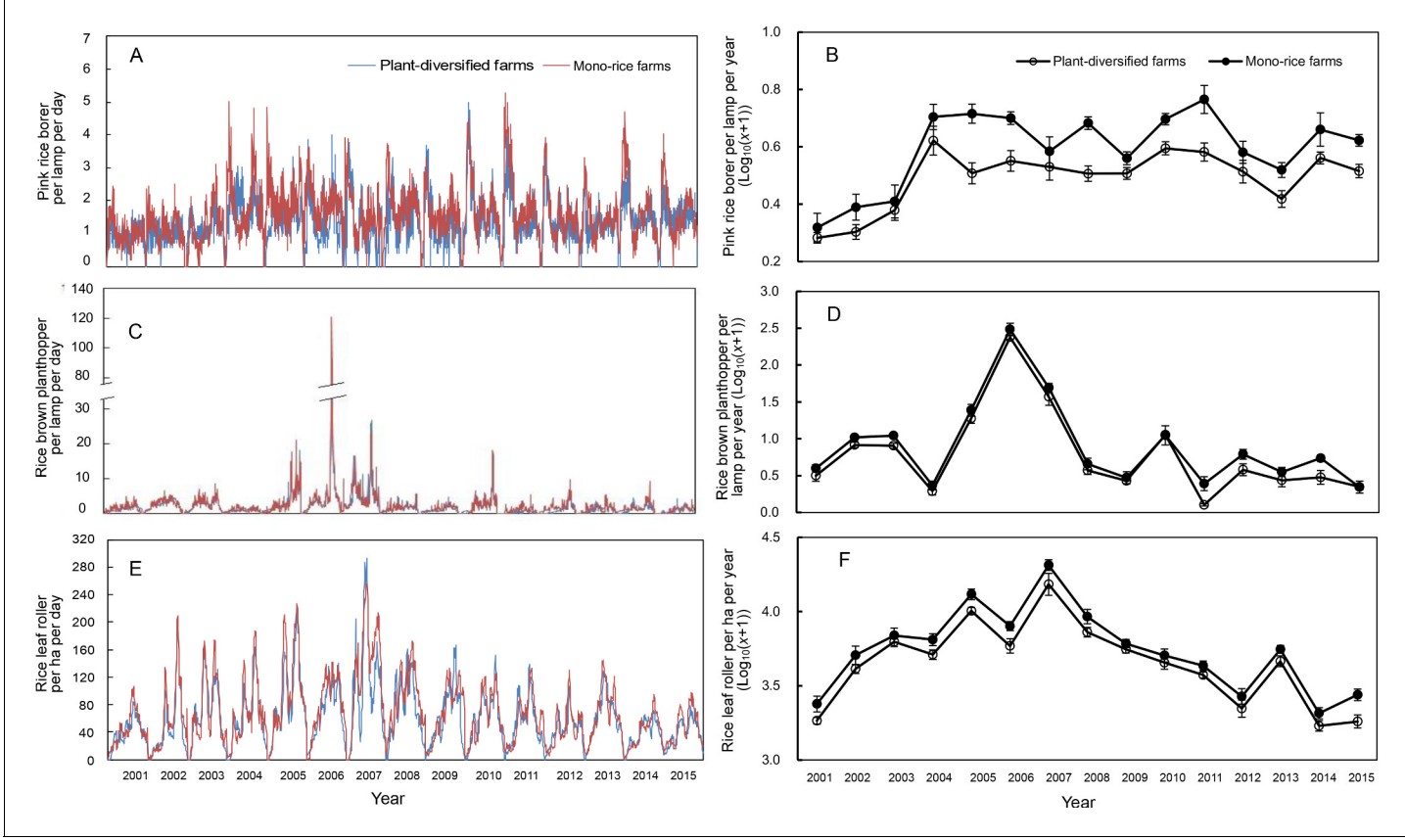

**Figure 2.** Population dynamics of pink rice borer and brown planthopper trapped in the lamp and rice leaf roller observed in rice fields on plant-diversified and mono-rice farms from 2001 to 2015. (**A**) and (**B**) Pink rice borer; (**C**) and (**D**) Rice brown planthopper; (**E**) and (**F**) Rice leaf roller. The blue and red lines (in *Figure 6A, C and E*) indicate the plant-diversified and mono-rice farms, respectively. Vertical bars on each point denote SE. From 2001 to 2015, the number of trapped pink rice borers and rice brown planthoppers, and the population densities of rice leaf rollers were monitored from 10 April to 30 September, 11 May to 30 September, and 11 June to 20 September, respectively.

DOI: https://doi.org/10.7554/eLife.35103.005

The following source data is available for figure 2:

**Source data 1.** Pink rice borer: mean and standard deviation (individual per lamp per year) from the 15 year monitoring data, stratified by year and farm type.
DOI: https://doi.org/10.7554/eLife.35103.006
**Source data 2.** Rice brown planthopper: mean and standard deviation (individual per lamp per year) from the 15 year monitoring data, stratified by year and farm type.
DOI: https://doi.org/10.7554/eLife.35103.007
**Source data 3.** Rice leaf roller: mean and standard deviation (individual per ha per year) from the 15 year monitoring data, stratified by year and farm type.
DOI: https://doi.org/10.7554/eLife.35103.008
**Source data 4.** Economic Injury Levels of pink rice borers, rice brown planthoppers and rice leaf rollers issued by Shanghai Agricultural Technology Extension and Service Center (SATESC).
DOI: https://doi.org/10.7554/eLife.35103.009

p=0.001; amount of active ingredient insecticide: LR = 307.33, p<0.001), so interactions were included in the statistical models. In the first two years (2001 and 2002), the number of insecticide sprays and the amount of insecticide sprays did not decrease on the plant-diversified farms, but in the following 13 years (2003–2015), there were significant decreases (number of insecticide sprays: mean percent decrease = 16.8, SD = 4.9, range = 11.8–26.2, IQR = 13.0–18.9; amount of commercial insecticide sprays: mean percent decrease = 18.5, SD = 4.1, range = 10.1–25.4, IQR = 17.3–21.2; amount of active ingredient insecticide sprays: mean percent decrease = 27.1, SD = 17.2, range = 5.0–66.2, IQR = 18.2–27.3) (*Figure 5A–B*; *Figure 5—source datas 1–2*).

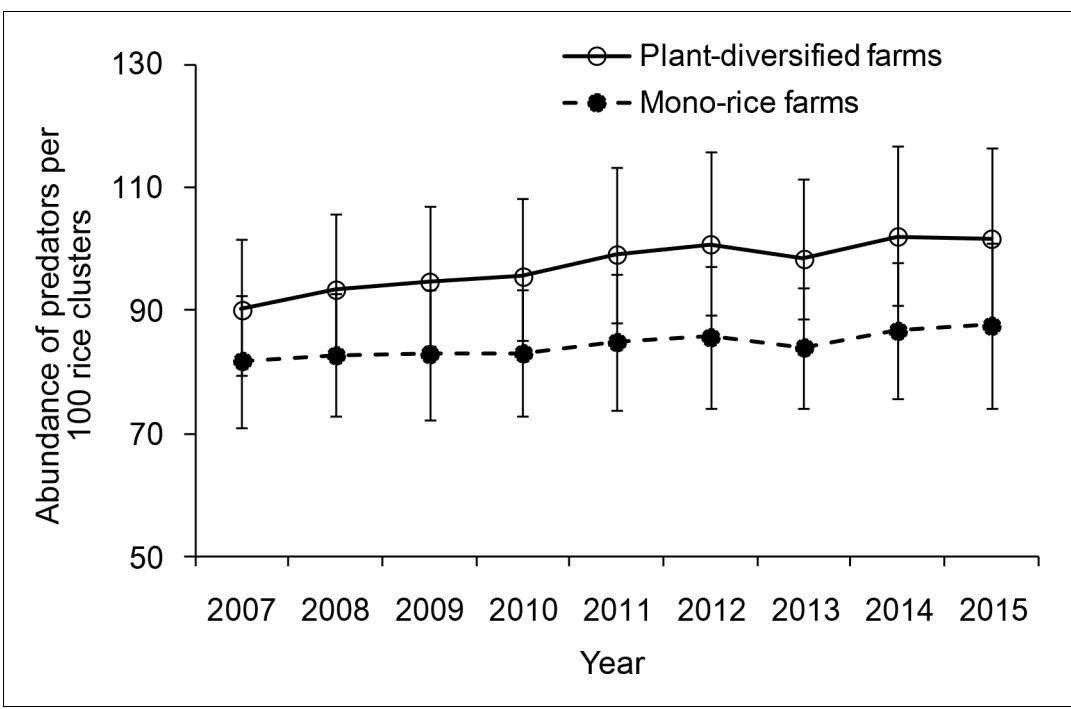

**Figure 3.** Abundance of predators observed on plant-diversified and mono-rice farms for each year. Abundance of predators in the rice fields was sampled at an interval of 15–20 days (with minor variation due to weather conditions) from seedling stage to grain stage from 2007 to 2015. Vertical bars on each point denote SE.
DOI: https://doi.org/10.7554/eLife.35103.010
The following source data is available for figure 3:

**Source data 1.** Predator abundances: mean and standard deviation (individual per 100 rice clusters per year) from the 15 year monitoring data, stratified by year and farm type.
DOI: https://doi.org/10.7554/eLife.35103.011

## Grain yield and economic cost–benefit analysis

Monitoring data for each year showed that plant-diversified farming generally had higher grain yield (*Figure 5C*). The farm type × year interaction effect for grain yield was marginally significant (LR = 27.214, p=0.058). According to the main-effects model with only farm type and year as predictors, the farm type effect was not significant (LR = 0.0005, p=0.991). The economic cost–benefit analysis showed that the plant-diversified farming generated a net advantage of 576 (±116) RMB per hectare per year, which was 3.48% (±0.79%) higher than the mono-rice farms (*Figure 5—source datas 3–4*).

## Common-location experiments

For the abundances of the three pest groups and the predators in the common-location experiments, the statistical models indicated that plant-diversified fields had significantly lower abundance of stem borer (LR = 7.343, p=0.043), and marginally significant lower abundance of rice plant-hopper and leaf roller (rice plant-hopper: LR = 6.946, p=0.066; leaf roller: LR = 5.792, p=0.084). The increase in predator abundance was also marginally significant (LR = 7.545, p=0.069). Averaged over both experiments, the numbers of the three pest groups all decreased during the four years (stem borer: mean percent decrease = 15.6, SD = 2.9, range = 12.4–18.8, IQR = 13.6–17.5; rice plant-hopper: mean percent decrease = 16.7, SD = 1.6, range = 15.2–18.8, IQR = 15.5–17.5; leaf roller: mean percent decrease = 16.0, SD = 3.2, range = 12.3–18.7, IQR = 13.9–18.5) (*Figure 6A–C*), and the abundances of their predators increased on the plant-diversified fields (mean percent increase = 12.7, SD = 2.6, range = 8.8–14.8, IQR = 12.2–14.0) (*Figure 6D*).

Significantly less insecticide was applied on the plant-diversified fields in the common-location experiments (LR = 7.818, p=0.048), and these had marginally significantly higher grain yield

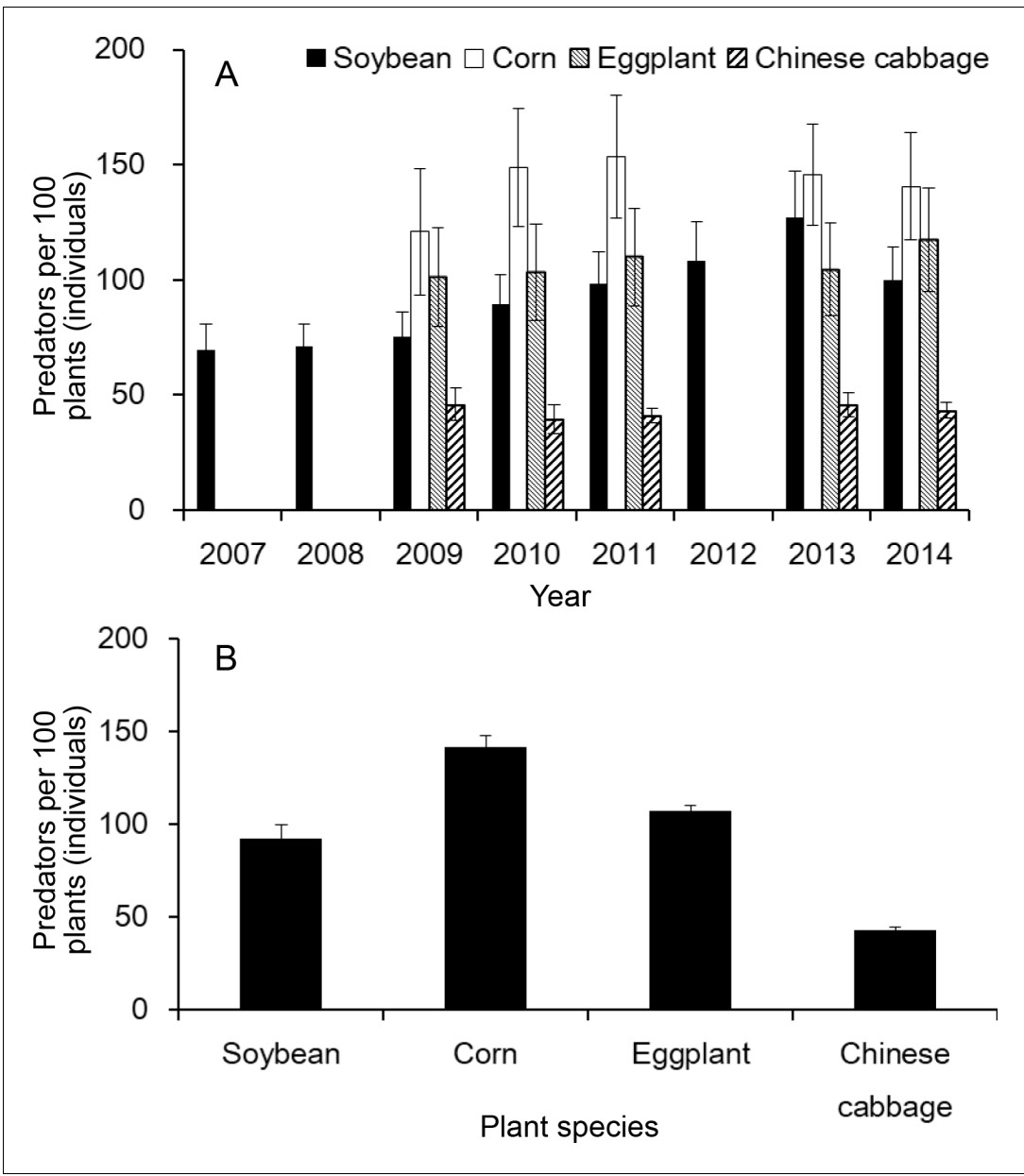

**Figure 4.** Predator abundance on the border crop of soybeans and neighboring crops (maize, eggplant and Chinese cabbage) on plant-diversified farms. (**A**) Predator abundance in different years; and (**B**) Predator abundance on four crops (soybean, maize, eggplant and Chinese cabbage). Abundance of predators (ladybird beetles, lacewings and spiders) were monitored in soybeans from 2007 to 2014 and in neighboring crops (maize, eggplant and Chinese cabbage) during 2009–2011 and 2013–2014. Vertical bars on each point denote SE.
DOI: https://doi.org/10.7554/eLife.35103.012

(LR = 6.691, p=0.054). Insecticides were applied at rate of 5.25 (±0.20)–7.16 (±0.15) kg•ha$^{-1}$ on the plant-diversified rice fields, which was, on average, 14.21% lower than the controls over the four years (SD = 4.07, range = 9.38–18.90, IQR = 11.94–16.65) (*Figure 6E*). Plant-diversified rice fields produced grain yields of 8.47 (±0.07)–8.73 (±0.05) t•ha$^{-1}$, which was on average 2.26% higher than the controls (SD = 0.28, range = 1.88–2.53, IQR = 2.13–2.45) (*Figure 6F*).

The power analyses suggested that by repeating the common-location experiment at an additional experimental site, the statistical power for achieving significant farm type effect on the abundances of rice plant-hopper, leaf roller, and predator could reach over 80%. For grain yield, 80% power for detecting significant farm type effect could be achieved with two additional experimental

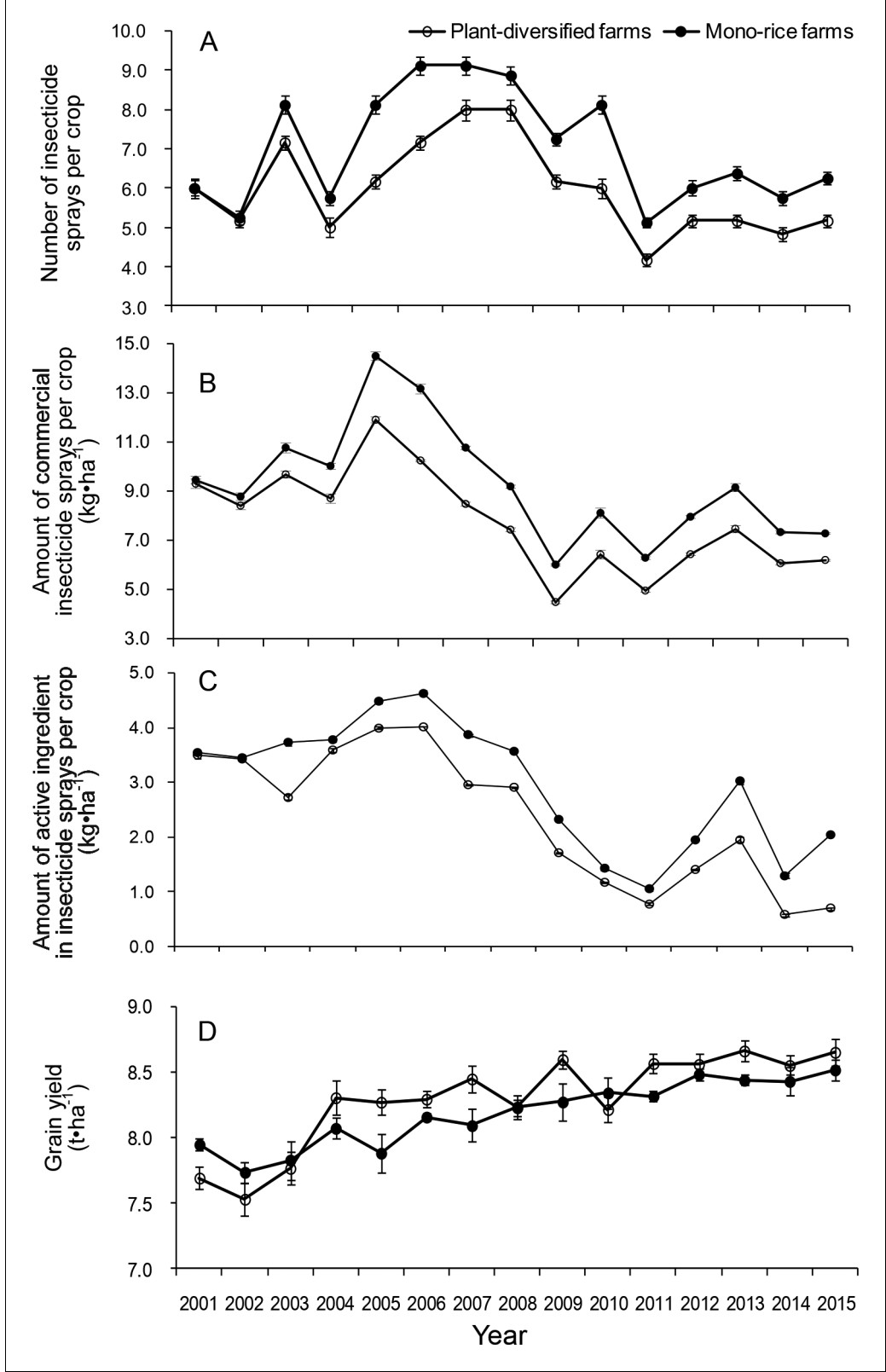

**Figure 5.** Dynamics of the number of insecticide sprays, the amount of insecticide sprayed and grain yield per crop on plant-diversified and mono-rice farms from 2001 to 2015. (**A**) Number of insecticide sprays; (**B**) Amount of commercial insecticide sprayed (kg•ha$^{-1}$); (**C**) Amount of active ingredient in insecticide sprayed (kg•ha$^{-1}$); and (**D**) Grain yield (t•ha$^{-1}$). Vertical bars on each point denote SE.

*Figure 5 continued on next page*

*Figure 5 continued*

DOI: https://doi.org/10.7554/eLife.35103.013

The following source data is available for figure 5:

**Source data 1.** The amount of commercial insecticide and active ingredient sprayed per crop: mean and standard deviation (kg•ha$^{-1}$) from the 15 year monitoring data, stratified by year and farm type.
DOI: https://doi.org/10.7554/eLife.35103.014
**Source data 2.** The number of insecticide spray per crop: mean and standard deviation from the 15 year monitoring data, stratified by year and farm type.
DOI: https://doi.org/10.7554/eLife.35103.015
**Source data 3.** Cost–benefit analysis of plant-diversified farms compared with mono-rice farms.
DOI: https://doi.org/10.7554/eLife.35103.016
**Source data 4.** Grain yield: mean and standard deviation (kg•ha$^{-1}$) from the 15 year monitoring data, stratified by year and farm type.
DOI: https://doi.org/10.7554/eLife.35103.017

sites. Adding years and replicates could also increase power, but not as effectively as increasing the number of sites; 80% power could not be reached even by large additions of years or replicates (*Figures 7–10*).

The abundances of the three predator groups (ladybird beetles, lacewings and spiders) were much higher in the maize neighbor crop (120.1 [±10.3]–146.3 [±4.3]) individuals per 100 plants) than in the soybean border crop (69.2 [±3.2]–93.1 [±7.4]) individuals per 100 plants) (*Figure 6—figure supplement 1*; *Figure 6—source datas 1–6*).

## Discussion

The initial establishment of monitoring sites of the community farms was initiated by farmers without knowledge of statistical design, so the establishment of 34 community farms was done without appropriate randomization in the 15 year study. While non-random distribution of the two farming types might have affected the results, it is unlikely that this could account for the main results. All 34 farms belong to Shanghai suburbs and have similar weather conditions and soil types. Because of the lack of randomization in the monitoring study, we performed the two common-location experiments, which verified the results of the monitoring study. Both the monitoring data on community farms and the results of the common-location experiments clearly indicated that increased predator abundance was concomitant with reduced pest abundance, resulting in reduced insecticide application on plant-diversified fields. In all likelihood, the reduced pest abundance contributed to rice growth, development and reproduction, and thus increased rice yield. These results were consistent across years in the monitoring study and in the common-location experiments. The reduction in insecticide application and decreased labor cost of spraying insecticides, together with the slightly higher grain yield on the plant-diversified farms, resulted in an increased economic benefit.

The increased abundance of predators in more diverse fields likely occurred because the higher diversity of plants provided 'resource pools' for the predators. The additional plants provided food sources (nectar, pollen, honey dew, etc.), as well as breeding habitats and refuges for these predators, allowing them to increase their longevity and fecundity (*Mason et al., 2014*; *Wan et al., 2016a*; *Woodcock and Heard, 2011*). A previous study indicated that these predators can disperse up to 40 meters, and can therefore move into rice fields from adjacent habitats and crops (*Yu et al., 2002*). Thus, the most likely explanation is that the increase in the abundance of invertebrate predators spread from border and neighboring crop areas onto the rice fields, increasing predation on pest populations. A similar effect due to neighboring crops was observed in Bt cotton fields (*Lu et al., 2012*). Diverse crops and weeds may also provide similar advantages for parasitoids, which have been shown to exhibit improved biocontrol services, reducing pests feeding on plant tissues and plant products, such as pollen, nectar, and sap (*Gurr et al., 2016*; *Wäckers et al., 2007*).

Ecosystem service providers and service-providing units are concepts that have been used to study ecological intensification (*Gurr et al., 2016*; *Luck et al., 2009*). In our study, diverse plants surrounding rice fields can be considered ecosystem service-providing organisms, supporting predators that reduced the populations of pest herbivores. The diverse plant communities initiated a trophic

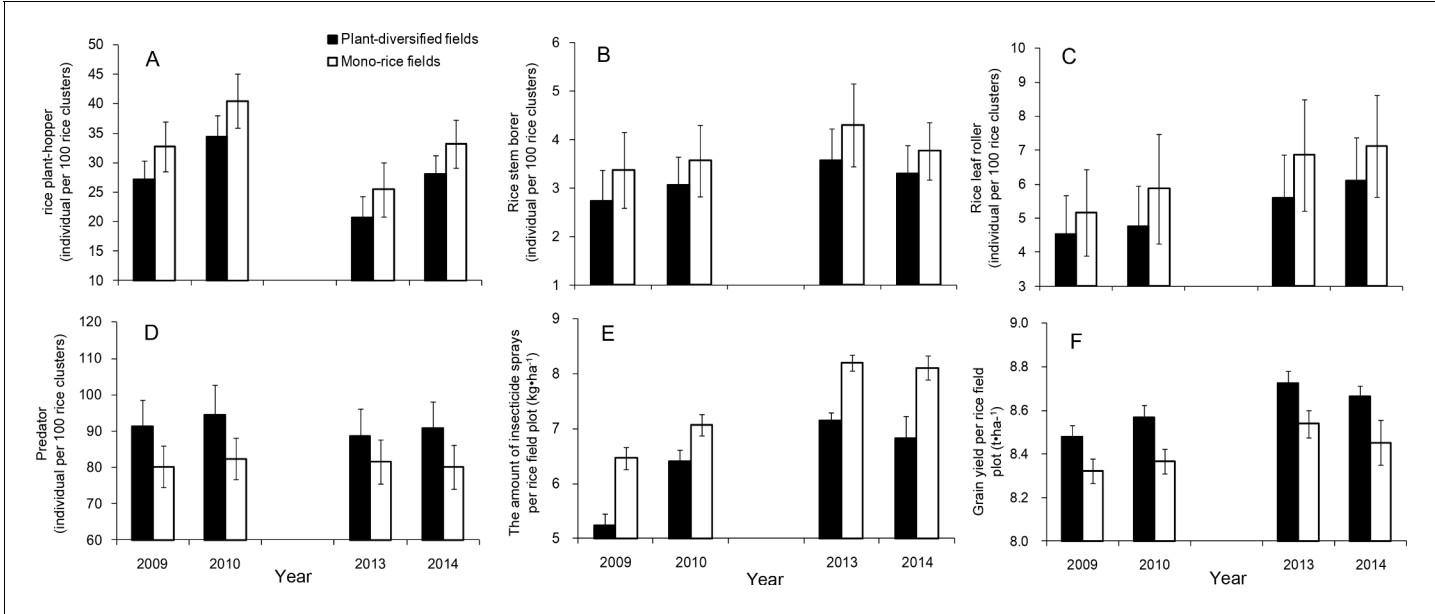

**Figure 6.** Effects of plant diversification in experiment in which plant-diversified with mono-rice farming were compared at the same locations. (A) Density of rice plant-hoppers sampled in rice field plots; (B) Density of rice stem borers sampled in rice field plots; (C) Density of rice leaf rollers sampled in rice field plots; (D) Density of the predators (ladybird beetles, lacewings and spiders) sampled in rice field plots; (E) Amount of insecticide sprays per rice field plot (kg•ha$^{-1}$); and (F) Grain yield per rice field (t•ha$^{-1}$). Vertical bars denote SE.

DOI: https://doi.org/10.7554/eLife.35103.018

The following source data and figure supplement are available for figure 6:

**Source data 1.** Stem borer: mean and standard deviation (individual per 100 rice clusters) from the common-location-experiments, stratified by year, farm identity, and farm type.
DOI: https://doi.org/10.7554/eLife.35103.020
**Source data 2.** Rice plant-hopper: mean and standard deviation (individual per 100 rice clusters) from the common-location-experiments, stratified by year, farm identity, and farm type.
DOI: https://doi.org/10.7554/eLife.35103.021
**Source data 3.** Leaf roller: mean and standard deviation (individual per 100 rice clusters) from the common-location-experiments, stratified by year, farm identity, and farm type.
DOI: https://doi.org/10.7554/eLife.35103.022
**Source data 4.** Predator: mean and standard deviation (individual per 100 rice clusters) from the common-location-experiments, stratified by year, farm identity, and farm type.
DOI: https://doi.org/10.7554/eLife.35103.023
**Source data 5.** Insecticide amount: mean and standard deviation (kg•ha$^{-1}$) from the common-location-experiments, stratified by year, farm identity, and farm type.
DOI: https://doi.org/10.7554/eLife.35103.024
**Source data 6.** Yield: mean and standard deviation (kg•ha$^{-1}$) from the common-location-experiments, stratified by year, farm identity, and farm type.
DOI: https://doi.org/10.7554/eLife.35103.025
**Figure supplement 1.** Predator abundance in the border crop of soybeans and in the neighboring crop of maize sampled in rice field plots.
DOI: https://doi.org/10.7554/eLife.35103.019

cascade in rice-herbivore-predator interactions, in which increased abundance of predators reduced their herbivores, which in turn resulted in a positive effect on the primary productivity and rice yields, permitting changes in management: a reduction of insecticide inputs and labor. These changes resulted in a small but immediate increase in economic performance (*Figure 11*). The benefits of plant-diversified farming have not previously been assessed on such a long-term basis. Some studies have reported that plant-diversified farming resulted in lower yields and higher pest densities, leading to criticism of the approach by some scientists and rejection by some farmers (*Dassou and Tixier, 2016*; *Letourneau et al., 2011*). Maintaining high crop yields is important if ecological intensification is to be widely accepted (*Cassman, 1999*; *Geertsema et al., 2016*; *Pywell et al.,*

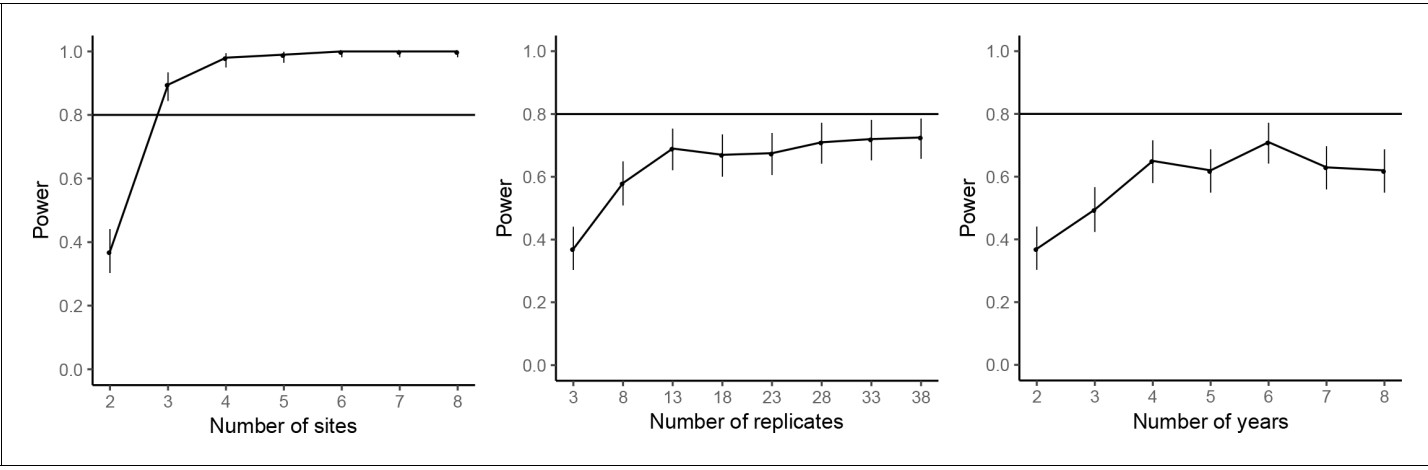

**Figure 7.** Power analysis results for rice plant-hopper occurrence, where the power are estimated for additional experimental sites (left), additional replicates for each farm type–site–year block (middle), and additional years (right), based on the effect size estimated by the mixed-effect model fitted to the common-location experiment data. The horizontal line marks the standard power level of 80%. Vertical line denotes the 95% confidence interval for the power estimates.

DOI: https://doi.org/10.7554/eLife.35103.026

*2015*; *Zimmerer, 2013*). In our study, we actually observed a slight increase in crop yield and economic performance, while the pesticide use was reduced, enhancing agricultural sustainability.

Organic agriculture has the potential to promote ecological intensification (*Crowder et al., 2010*; *Winqvist et al., 2012*), but its adoption has been limited, in large part because organic farming usually produces lower short-term yields (*Seufert et al., 2012*), although this can be compensated for by higher prices for the products in some markets. Conventional farming has been based on crop monocultures and large inputs of mineral fertilizers and pesticides, making it difficult to provide farmers with feasible strategies for incorporating ecological intensification. Biological diversification can be a first step in promoting ecological intensification of conventional agriculture (*Gurr et al., 2016*), as it often shows an immediate positive effect through the provisioning of ecosystem services (*Bommarco et al., 2013*), as demonstrated in our study.

Unlike smallholder farmers, urban farmers tend to be better educated and have a broader vision of farming. They have more contact with and obtain more information from policy makers and scientists, and are therefore more open to new ideas. Thus, ecological intensification of agriculture can be more easily implemented in urban than rural agriculture in China. Support from the government,

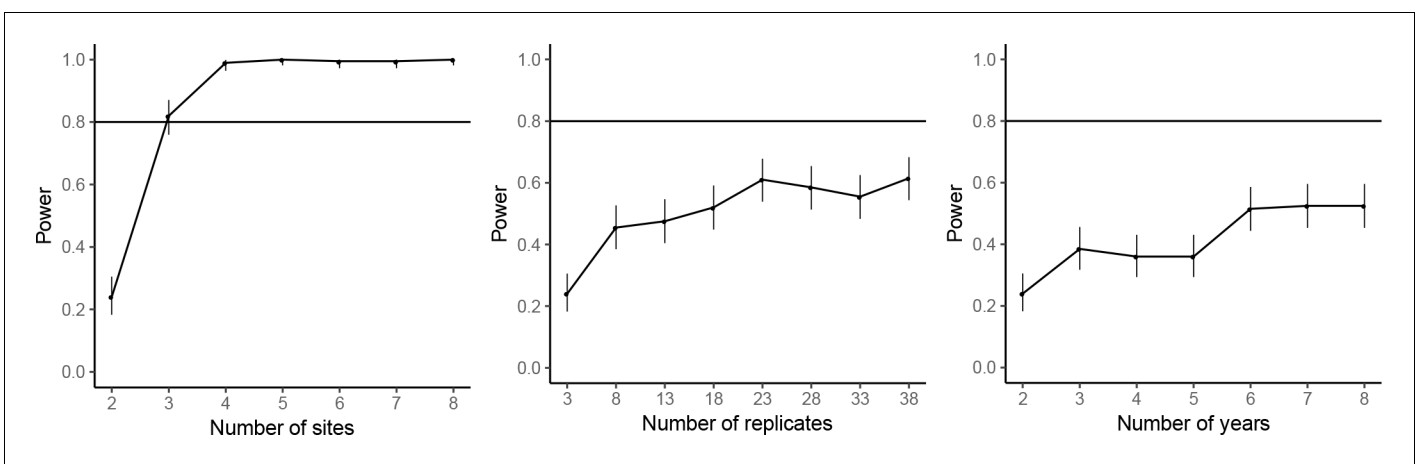

**Figure 8.** Power analysis results for rice leaf roller occurrence. See caption of Figure 7—figure supplement 1 for details.

DOI: https://doi.org/10.7554/eLife.35103.027

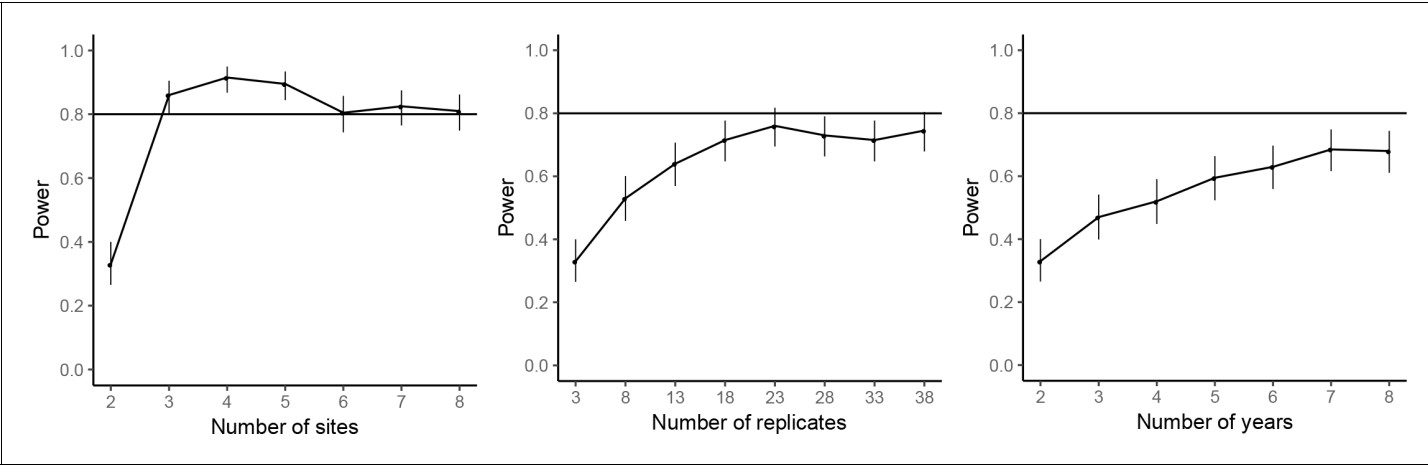

**Figure 9.** Power analysis results for predator abundance. See caption of Figure 7—figure supplement 1 for details.

DOI: https://doi.org/10.7554/eLife.35103.028

including new laws and regulations, can facilitate ecological intensification. It is encouraging that 93.8% of urban farmers would like to adopt plant-diversified farming to promote ecological intensification, according to a questionnaire survey (*Wan et al., 2016b*). There is reason to expect that plant-diversified farming will be practiced extensively in urban agriculture in the near future. One major challenge is the ongoing conversion of agricultural land, sometimes very high quality agricultural land, to housing, industry and commercial use throughout the world (*Singh and Swami, 2015*).

Our results on pests and pesticide application resulting from the introduction of border and neighboring crops, without a highly-developed theory or a vast amount of quantitative data on the relationship between plant diversity and ecosystem services, suggest that the potential for ecological intensification through increased plant diversity is much greater than what we have shown here. A major research effort in ecological intensification could result in revolutionary changes in agriculture, in which pesticide application can be reduced drastically or even eliminated, improving food safety and environmental quality off as well as on farms. Urban agriculture can serve a laboratory for such research, while providing an important source of food for city dwellers. We urge researchers to focus on forms of biotechnology that can contribute to ecological intensification, which has a great potential to conserve and promote biodiversity, while contributing to the provision of food to a growing human population (*Pywell et al., 2015*; *Zhao et al., 2016*; *Weiner, 2017*), rather than on forms of biotechnology that increase unsustainable intensification.

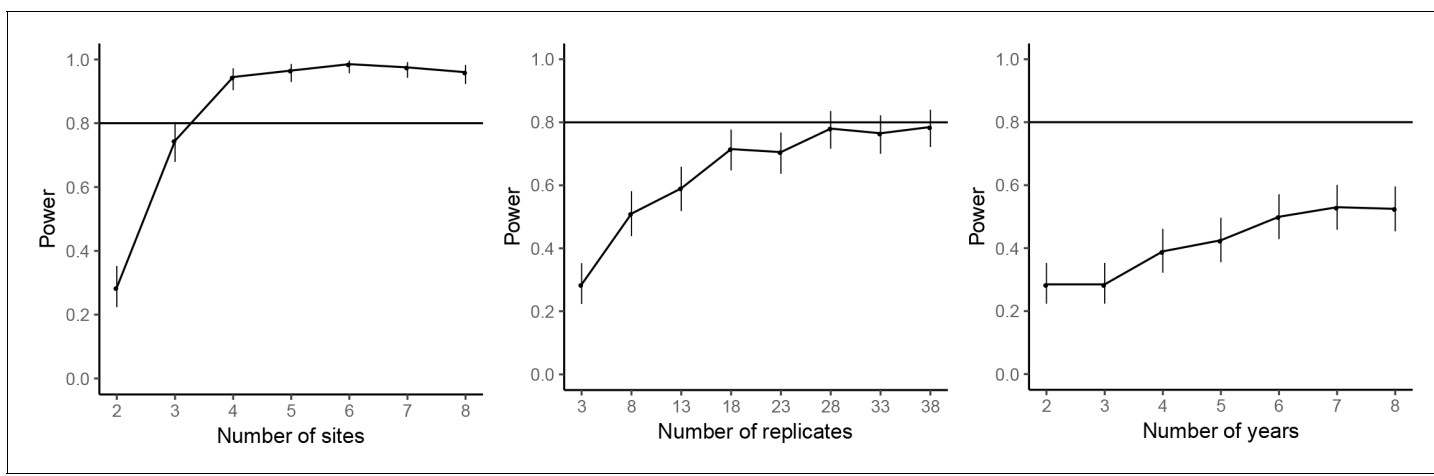

**Figure 10.** Power analysis results for grain yield. See caption of Figure 7—figure supplement 1 for details.

DOI: https://doi.org/10.7554/eLife.35103.029

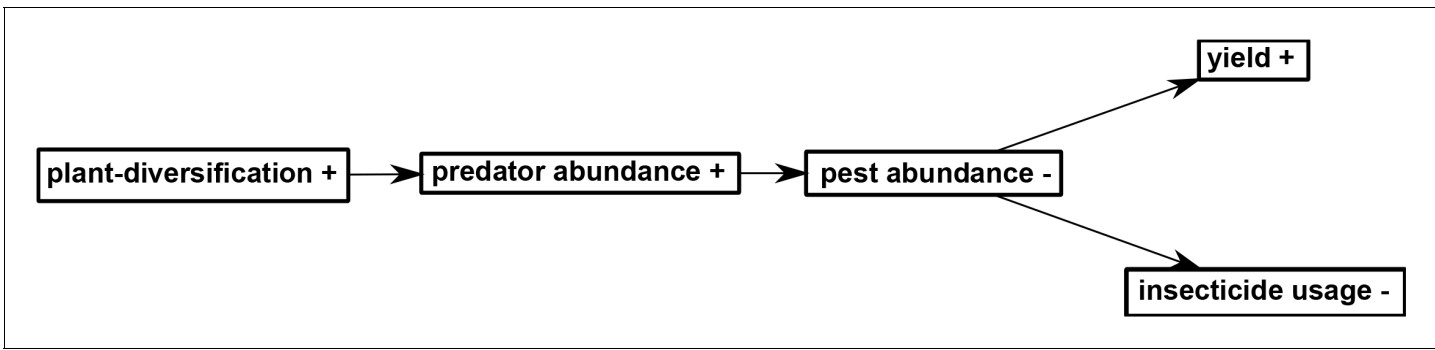

**Figure 11.** A path diagram depicting the hypothesis on the relationships among the variables considered in this study. '+' indicates increased level and '-' indicates decreased level.

DOI: https://doi.org/10.7554/eLife.35103.030

## Materials and methods

### Monitoring sites

In this study, all community farms are located in nine districts in the suburban areas of Shanghai, China, belonging to alluvial plains of the Yangtze River Delta (*Figure 1*). All sites are in the same geographic and climatic region (East Asian monsoon zone, with four distinctive seasons). Southeast winds prevail in summer, producing hot, rainy weather, while southerly winds dominate in winter, causing winter monsoons.

In early 1990s, SATESC established 28 community farms (120°54'–121°51'E, 30°47'–31°28'N) in eight districts (Minhang, Jiading, Baoshan, Pudong, Fengxian, Songjiang, Qingpu and Jinshan districts) to monitor pest occurrence in rice fields. As these 28 community farms were planted with the sole crop (rice), we consider them as the control farms. After repeated requests by the Chongming Agricultural Technology Extension and Service Center (CATESC), SATESC established another six community farms (121°12'–121°46'E, 31°32'–31°49'N) in Chongming district for monitoring pest occurrence in rice fields in late 1990 s. One to six monitoring farms with fixed staff were established in each district of Shanghai in late 1990s (*Figure 1*; *Figure 1—source data 1*). In Chongming district, the six community farms were labor intensive, and the farmers grew a border crop of soybeans around the periphery each rice field and common vegetable crops (maize, eggplant, Chinese cabbage, etc.) around the whole rice-growing area. Thus, we considered the six community farms as the treatment farms. Each monitoring farm in the control and treatment had an area of 1.3–2.0 hectares. We collected the data from the 34 community farms from 2001 to 2015.

The farmers within each community farm applied pesticides according to pest forecast information offered by the Plant Protection Station affiliated to the Agricultural Technology Extension and Service Center in each district from which we obtained information on the pesticide application from 2001 to 2015. The application of insecticides (time, type, amount, target crop, etc.) followed the 'Pest Control Guidance' issued by SATESC, so that insecticides used were the same on the treatment and control farms each year (*Figure 1—source data 2*). The managers of each farm were directed to apply insecticides only when pest abundance reached the Economic Injury Level, which is the basis for decision-making in integrated pest management (IPM) programs (*Higley and Pedigo, 1993*). Thus, the number of insecticide sprays and the amount of insecticide sprayed each year varied among the 34 community farms, because the abundance of pests varied among the farms.

We obtained the pest information for mono-rice and plant-diversified farms from the eight districts and from the six farms in Chongming district, respectively. The farmers in the 34 community farms uploaded their pest information through the server into the Shanghai Pest Monitoring System, which was accessible to the monitoring locations, 9 districts of Agricultural Technology Extension and Service Centers and SATESC. Occasionally, a few farmers on the 28 mono-rice farms in eight districts (Minhang, Jiading, Baoshan, Pudong, Fengxian, Songjiang, Qingpu and Jinshan districts), did not upload monitoring data for pests, so these data were not in the system when the SATESC released the pest information for each district.

On each farm, the rice growing area was divided into nine paddy plots, which were 60–70m × 25–35 m on each plot. There were 0.3–0.5 m wide earth banks around each paddy to retain water for the rice crop. On the control farms, these ridges were left bare. On the plant-diversified farms, a border crop of soybeans was hand-sown with hill-seeding on the bunds. Neighboring crops (maize, eggplant, Chinese cabbage, etc.) were intersown in an approximately 6–10 m wide area around the periphery of the whole area. Pesticides were not applied to the soybean borders or the neighboring crops on plant-diversified farms, nor was there any form of weed control in the soybean or neighboring crop areas. All the community farmers applied herbicides within the rice fields in both the mono-rice and plant-diversified farms.

On all community farms, the hybridized rice varieties were cultivated and agronomic practices were basically the same each year. At the end of the growing season, in a way that was consistent across all farms, the SATESC asked the technical staff of the 9 districts of Agricultural Technology Extension and Service Centers to measure the grain yield at all locations. Three to six rice field plots of 0.120–0.167 hectares were selected in each community farm to measure the grain yield each year. The fully mature rice plants were cut and threshed ('Z'-style sampling with 10, one square meter subplots in each plot) (*Figure 12*), and the grain yield per unit area determined.

## Monitoring and sampling methods

Insect trapping lamps (Jiaduo Company Limited, Henan Province, China) were installed on the periphery of 9 paddy plots of each community farm to estimate the abundance of the main herbivore pests (pink rice borer, rice brown planthopper, etc.). Lamps attracted the pests, which fell into cloth bags or cylindrical iron buckets below the lamp, and then the workers on each farm identified the species and counted the pests every day. Each lamp tube was hung 1.5 m above the ground and was automatically turned on at 6:00 p.m. and turned off at 5:00 a.m. from early April to late October (*Wan et al., 2016a*). From 2001 to 2015, we monitored the number of trapped pink rice borers and rice brown planthoppers from 10 April to 30 September and from 11 May to 30 September, respectively, as this was when they occurred in the rice fields.

During the emergence period of rice leaf rollers, a count of adult moths was taken from a sample plot of $66.7 \times 10^{-3}$ hectares around the lamp. A two-meter long bamboo pole was used to slowly shake the top half of rice plants, working upward, and the total number of flying moths observed in the sampling plot was considered an estimate of the density of adult rice leaf rollers (*Wan et al.,*

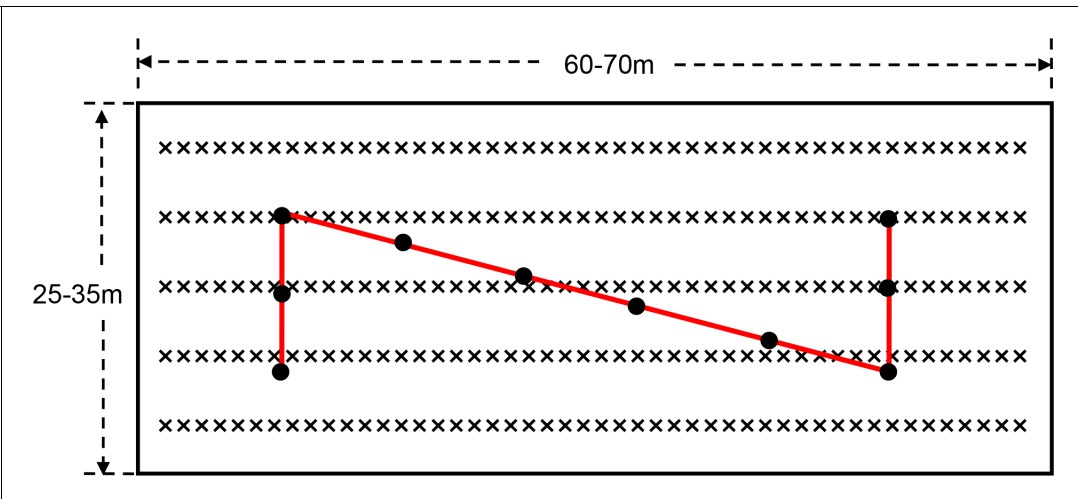

**Figure 12.** The layout of each rice plots in each community farm and 'Z'-style grain yield sampling. On each farm, the rice growing area was divided into nine paddy plots, which were 60–70 × 25–35 m on each plot. Three to six rice field plots of 0.120–0.167 hectares were selected in each community farm to measure the grain yield each year. The rice plants at harvest stage ('Z'-style sampling with 10, one square meter subplots in each plot) were mowed and threshed, and the grain yield per unit area was obtained. '×' and black solid dots denoted rice, and 1 m² sampling areas for rice grain yield, respectively. The interval between two adjacent black solid dots was about five meters.
DOI: https://doi.org/10.7554/eLife.35103.031

*2015*). From 2001 to 2015, the population densities of rice leaf rollers were sampled from 11 June to 20 September.

Monitoring is very labor intensive, so the abundance of the three major generalist predator groups (ladybird beetles, lacewings and spiders) could only be monitored on some of the community farms. Three to six plant-diversified farms and 6–10 mono-rice farms were selected to monitor the predators from 2007 to 2015 (*Figure 1—source data 1*). 'Z' style scouting using five subplots consisting of 20 adjacent clusters of rice plants was performed from the seedling to harvesting stage in the treatment and control farms, giving 100 clusters of rice plants. Each sampling was performed every 15–20 days (with some variation due to weather conditions) from 2007 to 2015. Over the whole rice growing period, there were nine sampling dates each year: two in the seedling, tillering and grain stages and three during the booting stage.

The population dynamics of the three major generalist predator groups (ladybird beetles, lacewings and spiders) was evaluated approximately every 15 days from middle/late June to early October in the border crop of soybeans on 4–6 plant-diversified farms from 2007 to 2014, and in the three main neighboring crops (maize, eggplant and Chinese cabbage) in 2009–2011 and 2013–2014, giving seven sampling dates each year for the predators in border and neighboring crops. At each sampling date, a subplot of 20 adjacent plants at five randomly selected locations were visually investigated and all predators counted and recorded (*Lu et al., 2012*).

## Common-location experiments

Because the mono-rice and diversified farms in the 15 year study were located in different suburban areas of Shanghai, we also performed two more highly controlled 2 year experiments in which both treatments were performed at the same locations. One experiment was conducted in Xinchang Town, Pudong district (121°38′E, 31°01′N), in 2009 and 2010 and the other in Sanxing Town, Chongming district, (121°17′E, 31°44′N) in 2013 and 2014. For each site, the experiment was a randomized block design with three replicate blocks. Each block contained two plots — one for plant-diversified farming and the other for mono-rice farming (50–55 × 30–35 m plot size in Xinchang, and 55–60 × 35–40 m in Sanxing) (*Figure 13*). In early-to-middle May of each year, a border crop (soybean) was planted on the earth banks around the plots of the diversified cropping treatments. In the periphery of each rice field plot of the diversified cropping treatments, a neighboring crop (maize) was interplanted in early June on the 1.5–2.0 m-wide bare soil border around soybeans. Blocks were spaced more than 100 m apart, and adjacent plots in each block were separated by a > 50 m wide buffer zone consisting of a rice field that was managed according to normal pest management, which was identical in all control treatments.

Hybridized rice variety 'Huayou 14' (bred by Shanghai Academy of Agricultural Sciences) was transplanted in the middle-to-late June each year, and agronomic practices such as soybean planting, weed management, pesticide use, irrigation, etc., were the same as on the monitoring sites. Three hundred kg nitrogen per hectare was used in each rice field plot, with 70% of this applied at the basal-tillering fertilizer and 30% at the panicle stage.

Workers sprayed the rice crop with pesticides over the course of the growing season as deemed necessary according to the 'Pest Control Guidance' issued by SATESC. The number of insecticide sprays and the amount of insecticide sprayed were based on the pest abundance in all experimental blocks. As in the monitoring study, the workers in each block were directed to withhold insecticide applications unless pest abundance reached the Economic Injury Level issued by SATESC. To ensure the continuity and reliability of the information, we kept written record of the insecticide input details for each plot.

The pests and predators in each plot of rice fields were scouted and sampled at 10–15 day intervals from the seedling to the harvesting stage (ten sampling dates each year), using methods similar to those described above, except, instead of using a trapping lamp, a white stainless 0.4 × 0.3 m steel plate was placed at the base of rice plants (*Cheng, 2001*). Immediately thereafter the rice plants were slapped with hands so that pests and predators (including lacewing larvae) fell into the plate. The numbers of pests and predators on the plate were then counted. To further survey the abundance of stem borers and leaf rollers in each plot, we examined infested rice stems and leaves at each sampling point and counted the stem borers and leaf rollers present. The abundance of rice stem borers or leaf rollers at each sampling location was the sum from the plate and from the infested stems or leaves.

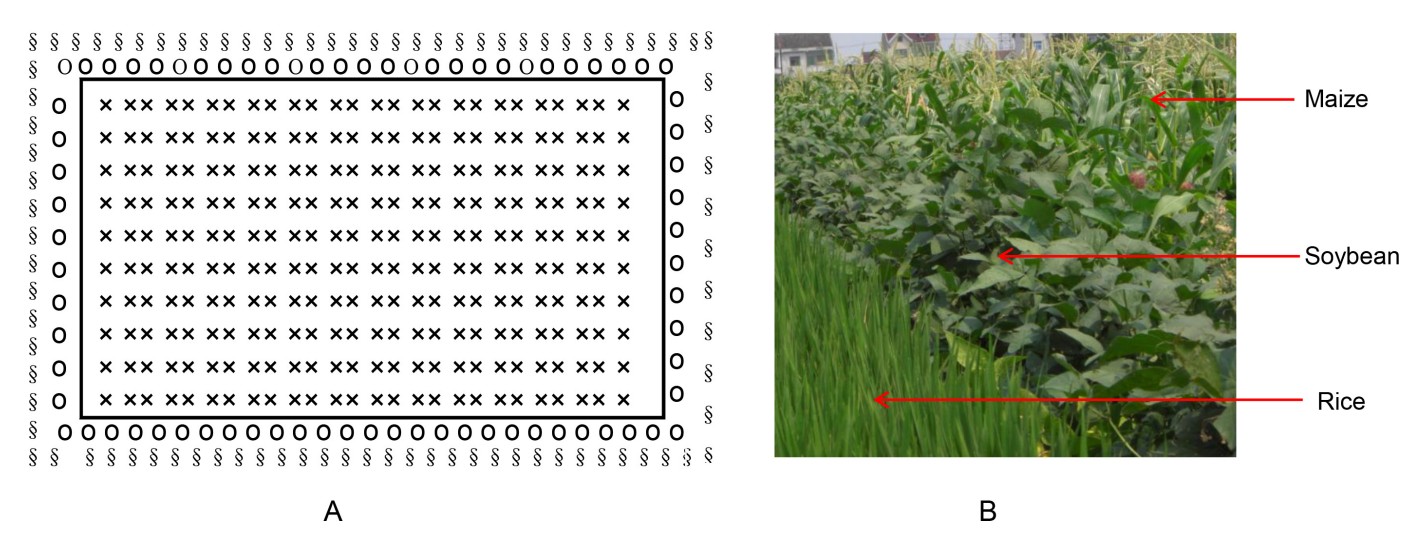

**Figure 13.** The layout of plant-diversified fields in which a border crop (soybean) was interplanted around the rice fields and a neighboring crop (maize) was interplanted around the soybeans. (**A**) Drawing plot diagram for plant-diversified fields; and (**B**) the photograph for plant-diversified fields. '×', 'o' and '§' denoted rice, soybean and maize, respectively. The layout of each control rice field was similar, but without soybeans or maize.
DOI: https://doi.org/10.7554/eLife.35103.032

The abundance of the three major generalist predator groups (ladybird beetles, lacewings and spiders) was surveyed about every 10 days in the border crop of soybean and in the neighboring crop of maize from middle-to-late June to late September in 2009 and 2010 in Xinchang, and in 2013 and 2014 in Sanxing, with ten sampling dates each year for the predators in soybean and maize plants in each plot in the plant-diversified field block. At each sampling date, 100 plants from five randomly-selected subplots of 20 adjacent plants were visually investigated and all predators were counted and recorded.

## Data analysis

The monitoring studies involved 34 farms belonging to 9 districts of Shanghai, and each farm had longitudinal repeated measurements across the 15 year study period. Due to the longitudinal nature of the data, we used mixed-effects models, a widely adopted method for analyzing data with repeated measurements (*Bolker et al., 2009*), to analyze the effects of the community farm types (plant-diversified vs. mono-rice farms) on the following eight response variables: numbers of pink rice borers and rice brown plant-hoppers trapped in the lamp, the population density of rice leaf roller observed in rice fields, the abundance of predators observed in the rice fields, the number of insecticide sprays, the amount of commercial and active ingredient insecticide sprayed, and grain yield. For predator abundance in the rice fields, the data used for analysis involved measurements from 6 to 10 mono-rice farms and 3–6 diversified-farms collected from 2007 to 2015. For the other seven outcome variables from 2001 to 2015, measurements for plant-diversified farming practices were obtained from the six community farms within Chongming district, and measurements from the 28 mono-rice farms were averaged at the eight district levels except for Chongming district (resulting in eight averaged measurements for mono-rice farming and six measurements for plant-diversified farming, collected annually across the 15 year monitoring period).

To account for the correlation among the outcomes at neighboring years within each farm identity, the mixed-effect model incorporates random intercepts and random coefficients for the polynomial year terms to model variation in yearly trends among the farms. Third-degree polynomials were used to model the yearly trends within each farm as the yearly trends were nonlinear for the majority of the responses (*Figures 2*, *3* and *5*). Altogether, year is treated as a categorical variable in the fixed-effect component of the model for adjustment of the overall year effect across all locations, and as a continuous variable in the random component for adjusting the within-farm yearly trends. To determine the significance of farm type, linear mixed effect models with fixed-effects farm type,

year, and interaction between farm type and year were first used to test whether the interaction effect was significant using model comparison tests. If the interaction effect was not significant, the interaction term was dropped from the model, and the significance of farm type determined using a model with only farm type and year as fixed-effects. If the interaction term was significant, the farm type effect was analyzed in conjunction with the interaction effect. The numbers of the three pest groups were $\log_{10}(X + 1)$ transformed to reduce the influence of a few outlying observations.

In the common-location experiments, replicated measurements were obtained from two farms, and each farm had measurements collected over two years. Here we also used a mixed-effects model to analyze the effects of the rice field types (plant-diversified fields and mono-rice fields) on the densities of rice plant-hoppers, stem borers, leaf rollers and predators sampled in rice field plots, the amount of insecticide sprays, and grain yield. In this mixed-effect model, the random-effects component contains varying intercepts, farm type effects, and year effects for the farm identities. The significance of the farm type effect was determined by model comparison tests, where the full model with the farm type factor was compared with the reduced model with the farm type factor removed.

Parametric bootstrap tests with 2500 bootstrap samples were used for significance testing (**Halekoh and Højsgaard, 2014**). The standard significance level 0.05 and a marginal significance level 0.10 were considered. R version 3.3.0 was used for the above analyses.

An economic cost–benefit analysis was conducted according to **Gurr et al. (2016)**, in which we considered the insecticide material costs in each treatment, the cost of seeds used to establish border and neighboring crops in the plant-diversified farming treatment. The analysis also reflected a labor cost for spraying insecticides, planting and soybean harvesting. The benefit was focused on the value of grain yield and the soybeans harvested on paddy field ridges. Other material inputs and labor costs were not included, as these were not consistent across treatments. The grain and soybean price per kilogram and the labor force was based on the average price in Shanghai each year. The economic costs of biodiversity loss, environmental pollution and human health caused by the negative effects of pesticides could not be included.

## Power analyses

For the common-location experiment data analysis, simulation-based power analyses were performed for marginally significant responses to farm type at the standard significance level of 0.05 for the abundances of rice plant-hoppers, leaf rollers and predators and for grain yield. Using the estimated effect size from the fitted mixed-effects model, the statistical power was estimated for number of experimental sites from 2 to 8, number of years from 2 to 8, and number of replicates within each farm type-site-year combination from 3 to 38. Statistical power was estimated based on 200 simulations. The target statistical power was 80%, and the R package 'SIMR' (**Green and MacLeod, 2016**) was used for power analyses. Due to the costs and manpower required, additional replication of the common-location experiments was not possible.

## Acknowledgements

We thank the agricultural technicians of the 34 surveyed community farms, the Shanghai Agricultural Technology Extension and Service Center and 9 districts of Agricultural Technology Extension and Service Center of Shanghai of China for collecting and providing data. We also thank Bernhard Schmid for helpful suggestions for the data analyses, and three anonymous reviewers for comments on the manuscript. This study was funded by grants from the Key Research Program of the Ministry of Science and Technology of China (2016YFD0200804), Shanghai Agriculture Commission of China (2017-1-2), Agriculture Research System of Shanghai, China (201703) and SAAS Program for Excellent Research Team (2018[B-01]).

# Additional information

## Funding

| Funder | Grant reference number | Author |
| --- | --- | --- |
| Key Research Program of the Ministry of Science and Technology | 2016YFD0200804 | Jie-Xian Jiang |
| SAAS Program for Excellent Research Team | 2018[B-01] | Jie-Xian Jiang |
| Agriculture Research System of Shanghai, China | 201703 | Nian-Feng Wan |
| Shanghai Agriculture Commission of China | 2017-1-2 | Jie-Xian Jiang |

The funders had no role in study design, data collection and interpretation, or the decision to submit the work for publication.

## Author contributions

Nian-Feng Wan, Conceptualization, Resources, Data curation, Software, Formal analysis, Supervision, Funding acquisition, Validation, Investigation, Visualization, Methodology, Writing—original draft, Project administration, Writing—review and editing, Performed the statistical model analysis; You-Ming Cai, Conceptualization, Resources, Data curation, Software, Formal analysis, Supervision, Funding acquisition, Validation, Visualization, Methodology, Writing—original draft, Project administration, Writing—review and editing; Yan-Jun Shen, Xiang-Yun Ji, Jun Li, Jie-Xian Jiang, Conceptualization, Resources, Data curation, Software, Formal analysis, Supervision, Funding acquisition, Validation, Investigation, Visualization, Methodology, Writing—original draft, Project administration, Writing—review and editing; Xiang-Wen Wu, Conceptualization, Resources, Data curation, Formal analysis, Supervision, Funding acquisition, Validation, Investigation, Visualization, Methodology, Writing—original draft, Project administration, Writing—review and editing; Xiang-Rong Zheng, Conceptualization, Data curation, Software, Formal analysis, Supervision, Validation, Methodology, Writing—original draft, Writing—review and editing, Performed the statistical model analysis; Wei Cheng, Conceptualization, Resources, Data curation, Software, Formal analysis, Supervision, Validation, Investigation, Visualization, Methodology, Writing—original draft, Writing—review and editing; Yao-Pei Jiang, Conceptualization, Resources, Data curation, Formal analysis, Supervision, Validation, Investigation, Visualization, Writing—original draft; Xin Chen, Conceptualization, Data curation, Software, Formal analysis, Supervision, Methodology, Writing—original draft, Writing—review and editing; Jacob Weiner, Conceptualization, Data curation, Software, Formal analysis, Supervision, Validation, Methodology, Writing—original draft, Writing—review and editing; Ming Nie, Software, Methodology, Writing—original draft; Rui-Ting Ju, Conceptualization, Data curation, Supervision, Validation, Methodology; Tao Yuan, Hao Zhang, Made the survey maps; Jian-Jun Tang, Conceptualization, Methodology; Wei-Dong Tian, Software, Methodology, Performed the statistical model analysis; Bo Li, Conceptualization, Resources, Data curation, Software, Formal analysis, Supervision, Validation, Visualization, Methodology, Writing—original draft, Writing—review and editing

## Author ORCIDs

Jacob Weiner http://orcid.org/0000-0002-0736-7943
Jie-Xian Jiang http://orcid.org/0000-0003-3567-8201
Bo Li https://orcid.org/0000-0002-0439-5666

## Decision letter and Author response

Decision letter https://doi.org/10.7554/eLife.35103.037
Author response https://doi.org/10.7554/eLife.35103.038

# Additional files

## Supplementary files
• Transparent reporting form
DOI: https://doi.org/10.7554/eLife.35103.033

## Data availability
All data generated or analysed during this study are included in the manuscript and supporting files.

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
