## [Decision Letter]

[Editors’ note: a previous version of this study was rejected after peer review, but the authors submitted for reconsideration. The first decision letter after peer review is shown below.]

Thank you for submitting your work entitled "Plant diversification contributes to ecological intensification of mega-urban agriculture" for consideration by *eLife*. Your article has been evaluated by a Senior Editor and three reviewers, one of whom, Bernhard Schmid (Reviewer #1), is a member of our Board of Reviewing Editors.

Our decision has been reached after consultation between the reviewers. Based on these discussions and the individual reviews below, we regret to inform you that your work will not be considered further for publication in *eLife*.

As mentioned at the pre-submission stage, we found your study potentially very relevant and interesting for a broad readership because it seems to show that farm-level plant diversity may increase crop protection by encouraging predators of pests such that less chemical pest control will be necessary without yield losses.

However, our problems with the full submission are still the same as with the pre-submission, namely that you do not clearly enough explain the lack of randomization in the 15-year study and you continue to use inappropriate, simple statistics. These problems unfortunately were seen by all reviewers, who in addition point out further issues, in part related to the lack of appropriate analysis. In particular, one reviewer mentions the possibility that the causality between reduced insecticide use and increased predator level could go either direction.

I hope that with the extensive comments of the reviewers you can further improve your manuscript for submission to another journal.

*Reviewer #1:*

As mentioned at the pre-submission stage, I find this study potentially very relevant and interesting for a broad readership because it shows that farm-level plant diversity may increase crop protection by encouraging predators of pests such that less chemical pest control will be necessary without yield losses.

My problems with the full submission remain, namely that the authors do not clearly enough explain the lack of randomization in the 15-year study (was it their choice not to randomize or was it because they met a "natural experiment" that had been started by others with lack of knowledge about statistical design) and use inappropriate, simple statistics. The first problem may be resolved by explaining in more detail *who* prepared the study and *why* it was not randomized. The second problem can be easily resolved by requesting the help of someone who knows how to properly analyze data from such experiments as the authors carried out.

One possible statistical model for the 15-year study is:

y ~ farm type + farm identity within farm type

+ year linear contrast + year as factor

+ farm type x linear year + farm identity x linear year

+ farm type x factorial year

Here "farm identity within farm type" is a random-effects term and *must* be included in the model to serve as an error term for "farm type" (which is now tested wrongly against the residual). Similarly, "farm identity x linear year" can be used as error term to test "farm type x linear year" if the authors decide to make this probably useful linear contrast.

It seems the authors also used t-tests in which they used the year-averages of y for the two farm types. However, this is not allowed because those are repeated measures of the same set of control and treatment farms. It is like comparing the height of a single tree given nutrients with that of a single control tree, using 15 years of growth data as replicates to test if there are significant effects of nutrients on tree height in more than the two particular tree individuals.

The same problem occurs with the randomized experiment, where the paired-sample t-test is again wrongly applied and the 2-way Anova too simplistic because it ignores "hierarchical structure", i.e. farm identity as random-effects factor. Here a possible statistical model is:

y ~ site + farm type + farm identity within farm type

+ year + farm type x year

Also, several speculative statements in the Discussion could be supported with more sophisticated statistical analysis of the data. In particular, the authors should use the fact that they have evaluated different variables at the same farms to calculate multiple linear regressions or include co-variates in the above models and perhaps integrate these analyses all into a structural equation model, that will allow the authors to explore their hypotheses how the different variables affected each other in a path-analytic diagram. At present these are speculations, even though the authors often present them as facts in the Discussion.

*Reviewer #2:*

Nian-Feng Wan and colleagues studied the impact of diversified rice cropping on pests, their predators and yield (including economic benefit) in 34 community farms in Shanghai, China. They found that rice fields with ridges planted by soybean and guards surrounding the crop area planted by maize, eggplant and Chinese cabbage, i.e. diversified crop fields, showed reduced pest infestation, higher predator abundance, less pesticide use and occasionally even higher yield. Economically, the diversified crop fields where more beneficial than conventional ones.

The study is based on an impressive dataset, including over 10 years of data from all the community farms, with high temporal and spatial resolution. Furthermore, due to the non-random distribution of the two farming types in Shanghai, the authors also performed a well-designed experiment to test if the results obtained from the long-term farm observations hold under experimentally controlled conditions. I therefore think that this is a very well conducted and extremely well-written manuscript. I enjoyed reading and I think this could be of interest to a wide audience of researchers and stakeholders, and therefore be potentially of high impact.

My main concerns are related to (1) the selection of the farms, (2) the set-up of the diversified farms, and (3) the insecticide use.

1) It is not clear to me how the farms were selected for this study. Some more information on the selection procedure would be highly appreciated. The sampling design is very unbalanced (28 conventional vs. 6 diversified farms) and not well distributed across Shanghai (the 6 diversified farms were all located on Chongming Island whereas none of the 28 conventional farms was located in this suburb). I guess there must be a good reason for this, which would clarify this slightly awkward sampling design.

2) It is unclear to me, how the 6 diversified farms were selected, in particular if these farms were set-up as such through an initiative by those farmers before the observations started or if they were set up specifically for this study through initiative of the researchers. In the latter case, the selection would be highly questionable and could potentially confound most of the results.

3) The results of the insecticide use (in particular the number of sprays) seem awkward, given that farmers "applied pesticides according to pest forecast information offered by the Plant Protection Station". I guess this forecast is the same for all farmers and would therefore suggest the same application regime for all farms. I therefore wonder to what extent the differences in pesticide use are simply caused by the contrasting interests of the different farmer communities. A proper pesticide use study would rather apply similar amounts of pesticide and assess their impact on the pest or reduce the pest to similar amounts and quantify pesticide use. However, in this study it seems that both pesticide use and pest abundance varied among farming types, resulting in difficulties to infer sound conclusions.

*Reviewer #3:*

The study analyses a long-term dataset of functional arthropod groups of herbivores and predators (herbivores: three different species of specialist herbivores on rice crop; predators: three generalist predators groups, namely ladybird beetles, lacewings and spiders), replicated over multiple farms and several years. In addition, they have data on insecticide use and yield. The core idea is to test whether a diversified farming system leads to an increased top-down control compared to a monoculture situation, followed by a reduced insecticide use and increased yield. They analysed a total of 6 diversified farms and compared it to 28 control farms. One farm was approximately 2 ha in size. The diversified farm consisted of 9 rice fields surrounded by a stripe of soybeans and an adjacent stripe of another crop. Accordingly, the monoculture farms had 9 rice fields which were surrounded by bare ground. I agree with the authors that we have a lack of knowledge on factors driving ecological intensification in urban systems. However, I have a couple of major concerns with the study.

First, the authors claim that the reduced insecticide use on diversified farms was a consequence of the increased top-down control due to increased predator densities. I wonder how they can rule out that it is not the opposite mechanism, namely that there was a reduced insecticide use on the diversified farms and as a consequence there was an increased predator density which then led to increased top-down control? In other words, how can they rule out that the result they find is not simply a consequence of reduced insecticide use but an effect of having multiple crops on a farm (i.e. diversified farm)?

A second major concern is the statistical approach used. First, the data are clearly time-series with nicely oscillating densities indicating lag-effects (Figure 2). In addition to lag-effects, the models should include more variables. For example, for explaining predator densities it would be important to incorporate the prey densities in the model. Second, the model (no matter whether it is a time-series analysis or an ANOVA) needs to incorporate the fact that repeated measures were obtained from the same location. Also, to analyse long-term data with multiple t-tests (for each year one test) is not appropriate. Third, the description of the statistical methods is confusing and incomplete.

A third major concern is that the main conclusion of higher predator control is not inferred from an experiment (for example, by exposing a standardized number of prey to predators) but based on a correlation.

Finally, the Introduction should be more put into a theoretical context, i.e. why and how do diversified farming systems drive biodiversity and ecosystem functioning. Further, the quality of the literature cited should be improved and all literature cited should be included in the reference list, e.g. Tscharntke et al., 2005.

[Editors’ note: what now follows is the decision letter after the authors submitted for further consideration.]

Thank you for resubmitting your work entitled "Plant diversity promotes biocontrol services, reduces insecticide use and increases rice yield in urban agriculture" for further consideration at *eLife*. Your revised article has been evaluated by Ian Baldwin (Senior editor), a Reviewing editor, and two reviewers.

The Reviewing editor and the reviewer who have seen the previous manuscript agree that the new submission has been improved. It was difficult to find a replacement for one of the previous reviewers but we did get a review of an expert with a similar research area, i.e. with knowledge about the more zoological aspects of your study. This new reviewer has similar concerns to the previous zoological reviewer about lack of detail in the methods description. We hope that you can add all the requested detail in a revision. Please also check the further comments by the Reviewing Editor and the previous reviewer as outlined below.

*Reviewing Editor (Reviewer #1):*

This resubmission of a previously rejected manuscript has been drastically improved by considering the reviewer comments and obtaining professional statistical help from two additional co-authors. Thus, the main message of the paper, that diverse margins around rice fields can reduce pests, increase predators, reduce insecticide use, marginally increase rice yield and increase economic benefit is now well supported by data and analysis. This is a very important contribution to the new topic of "ecological intensification" in the new field of "urban agriculture".

I have no further major comments. It would be good if co-author Jacob Weiner could have a very careful look at the manuscript before the revised version is uploaded again.

*Reviewer #2:*

The revised version of this manuscript is clearly improved and, in my opinion, the authors satisfactorily addressed the previous concerns by all the reviewers. I think this is an interesting study on the benefits of diversification in urban agriculture and shows results of an impressive amount of data. The design of the observational study is not perfect, but the authors managed to alleviate my main concerns with this. I therefore have no further major comments.

*Reviewer #3:*

General comments:

The topic of the paper is very important. The paper basically states that if plants (in fact, other crops) are planted around rice fields on the dams surrounding the fields, this reduces the need to use insecticides.

I concur with the reviews on the first version of the manuscript – the details of the study, in particular the experimental design and the statistical analysis were unclear. I did not assess the statistical analysis in detail.

Main points:

1) I find the title of the paper misleading. The manuscript does not analyse if increasing plant species richness increases biocontrol services etc., because the authors did not analyse diversity in the surroundings of the rice fields. Basically, they show that going away from the monoculture can be beneficial. Thus, the title should be "diversifying agriculture…" or "planting border crops enhances…"

2) Nowhere in the Results are a test statistic or the number of replicates are given.

3) Materials and methods: it is unclear how information on insecticide use was obtained.

4) Materials and methods: unfortunately, I find that still many details are unclear in the revision, see below.

5) Results: the effect on yield is only marginally significant and should not be overstated. Also, the authors provide no mechanism why yield was higher – do the insecticides not work?

Materials and methods:

Subsection “Monitoring sites”: the entire section should be restructured, the design of the diversified fields is described twice (first and third paragraphs), the same is true for other information. Start with the region, then describe the farms, then how rice is grown and then the different treatments.

Subsection “Monitoring sites”, first paragraph: please provide a drawing of a diversified field, and a photograph, how large is the area where other crops are grown.

Subsection “Monitoring sites”, second paragraph: the pest control guidance by SATESC is not known to the reader – what are the insecticides used, what active ingredients etc.?

Subsection “Monitoring sites”, last paragraph: what is a "Z" subplot? Again, use a drawing.

Subsection “Common location experiments”, first paragraph: design unclear: what is the replicate – a block, i.e. three pairs of?

Subsection “Common location experiments”, fourth paragraph etc.: Please provide more detail of the pest assessments: I assume the light kills insects attracted to the light? Were they collected in alcohol? Who identified the species? Why was the interval for the planthopper different if the same lamp in the same field is used? When you say each rice field, do you mean one in each of 9 plots per farm, or just one?

Subsection “Monitoring and sampling methods”, last paragraph: did you only count predators on plants?

Subsection “Common location experiments”, fourth paragraph: what is the correlation between number of pests and number of predators? I assume it is positive? Did you also count lacewing larvae?

I also do not understand how the farmers decided when to spray – you state that there are economic thresholds, so there must be data on how often the threshold was reached? Were the pest data used for the threshold analysis the same that you used?

Figures:

Figure 1: I thought the 28 farms were mono-rice and only 6 diversified, not the other way round?

Figure 2: were pest insects sampled all year round, or only from June onwards? This is not stated in the manuscript. In the first paragraph of the subsection “Monitoring and sampling methods” it states sampling was done May to September (April to September).

Figure 5: 9-14 kg insecticide per hectare is a lot – what is this number referring to, the liquid that is brought to the field? Details on the insecticides should be given and the amount plotted in active ingredient rather than some other unit.

---

## [Author Response]

[Editors’ note: the author responses to the first round of peer review follow.]

We thank the editors for their comments and suggestions, which helped us greatly to improve the quality of the manuscript. We have carefully studied the reviews and revised our manuscript accordingly.

Reviewer #1:

As mentioned at the pre-submission stage, I find this study potentially very relevant and interesting for a broad readership because it shows that farm-level plant diversity may increase crop protection by encouraging predators of pests such that less chemical pest control will be necessary without yield losses.My problems with the full submission remain, namely that the authors do not clearly enough explain the lack of randomization in the 15-year study (was it their choice not to randomize or was it because they met a "natural experiment" that had been started by others with lack of knowledge about statistical design) and use inappropriate, simple statistics. The first problem may be resolved by explaining in more detail who prepared the study and why it was not randomized.

We thank the editor for these important comments. We now explain fully the origin and background of the study, and the nature of the data. The answers to the question are found in the revised manuscript:

“To test our hypothesis, we refer to monitoring data collected from 34 community farms in 9 districts in suburbs of Shanghai, and we conducted two more controlled experiments for further verification. […] Since the diverse and mono-rice farms were located in different areas of the city, group members from the Shanghai Academy of Agricultural Sciences and Fudan University supplemented the monitoring data with two controlled experiments, in which both treatments were performed at two locations over two years in a complete random design.”

The second problem can be easily resolved by requesting the help of someone who knows how to properly analyze data from such experiments as the authors carried out.One possible statistical model for the 15-year study is:y ~ farm type + farm identity within farm type+ year linear contrast + year as factor+ farm type x linear year + farm identity x linear year+ farm type x factorial yearHere "farm identity within farm type" is a random-effects term and must be included in the model to serve as an error term for "farm type" (which is now tested wrongly against the residual). Similarly, "farm identity x linear year" can be used as error term to test "farm type x linear year" if the authors decide to make this probably useful linear contrast.It seems the authors also used t-tests in which they used the year-averages of y for the two farm types. However, this is not allowed because those are repeated measures of the same set of control and treatment farms. It is like comparing the height of a single tree given nutrients with that of a single control tree, using 15 years of growth data as replicates to test if there are significant effects of nutrients on tree height in more than the two particular tree individuals.

The model suggested by the reviewer has formed the basis of the re-analyses we had carried out. An appropriate statistical model for testing the farm-type effect should account for two design aspects of the 15-year monitoring studies: that farm identity is nested within farm type, and that there are longitudinal repeated measurements within each farm identity across the 15-year study period. To this end, we have re-analyzed the monitoring data based on the following model:

yijk=β0+β1farmTypei+β2yearFactorialj+β12farmTypei×yearFactorialj+μijk

μijk=b0,ik+b1,ikyearj+b2,ikyearj2+b3,ikyearj3+ϵijk,

i nested within farm type i. Finally, ϵijk is the random noise term for each individual measurement. Note that while year is a continuous term in the random-effects component, year is treated as a categorical factor (yearFactorial) in the fixed-effect component of the model.

We have applied a 3rd-degree polynomial transformation for the year variable as we have observed non-linear yearly trends among the various endpoints considered. This leads to a mixed-effect model with varying intercepts and varying coefficients for the polynomial year terms.

In the random-effects component term μijkfrom the model above, the random intercept term and the polynomial year terms are used to model the farm-specific effects and the farm-specific yearly trends. Together with the individual noise term ϵijk, the random component μijk is the error term used for testing the farm type x year interaction effect and the farm type main effect in the mixed-effects model.

To determine the significance of the farm type effect and the farm type x year interaction effect, we have employed a model comparison test, where the full model above was first compared against the model without the interaction terms. If the interaction effect was not significant, the interaction terms were removed and we then tested whether the farm type factor can be further removed from the model. For the test of significance, we have chosen the parametric bootstrap tests for mixed-effect model comparison. This test is preferable for small sample and/or unbalanced data, and was chosen here as the 15-year monitoring data was unbalanced (6 plant-diversified locations and 8 mono-rice locations).

A brief description of the above model has been added to the revised article (subsection “Data analysis”, first paragraph).

The use of the bootstrap test is also described in the revised manuscript (subsection “Data analysis”, third paragraph).

The same problem occurs with the randomized experiment, where the paired-sample t-test is again wrongly applied and the 2-way Anova too simplistic because it ignores "hierarchical structure", i.e. farm identity as random-effects factor. Here a possible statistical model is:y ~ site + farm type + farm identity within farm type+ year + farm type x year

Once again we thank the reviewer for pointing out the issues and the suggested model refinement. We have devised another mixed-effects model tailored for assessing the significance of the farm type effect while accounting for the design aspects of the common-location experiments. In detail, the common-location experiments involve two farms, with two consecutive years nested within each farm. For each experimental year, both farms contained two treatments, and three replicated measurements were retrieved from each treatment. This design inspired a model where the random-effects component contains varying intercepts, farm type effects, year effects, and farm type x year effects for the farm identities. Specifically, the model is:

yijkl=β0+β1farmTypei+μijkl

μijkl=b0,k+b1,kfarmTypei+b2,kyerj+ϵijkl,

where yijkl is a response variable, i∈ (mono-rice, plant-diversified) indicates the farm type, j∈ (first year, second year) specifies the year, k∈ (1, 2) indicates the farm identity, and l∈ (1, 2, 3) represents the replicates. μijkl is the random-effects component of the model, where b0,k, b1,ik, and, b2,jk are respectively the intercept, farm type effect, and year effect coefficients for farm identity k. Furthermore, ϵijkl is the noise term for the individual measurements. For the test of significance, we have chosen the parametric bootstrap tests for mixed-effect model comparison. This test is preferable for small sample and/or unbalanced data, and is chosen here due to the small sample size.

The revised manuscript now contains a brief description of the model above.

Also, several speculative statements in the Discussion could be supported with more sophisticated statistical analysis of the data. In particular, the authors should use the fact that they have evaluated different variables at the same farms to calculate multiple linear regressions or include co-variates in the above models and perhaps integrate these analyses all into a structural equation model, that will allow the authors to explore their hypotheses how the different variables affected each other in a path-analytic diagram. At present these are speculations, even though the authors often present them as facts in the Discussion.

Thanks for the pertinent suggestions and useful questions. In the Discussion, we have hypothesized that plant-diversification farming can increase predator abundance. The increased abundance leads to increased predation and thus decreased pest abundance. The decreased pest abundance further leads to reduced insecticide usage and increased grain yield. A possible path-analytic diagram describing the above hypothesis is shown in Figure 11—figure supplement 1 where “+” indicates increase and “- “indicates decrease. Through the use of structural equation modeling techniques, the fitness of the above model can be assessed, and alternative models could also be compared. Currently, however, the predator abundance data were not well matched with the other variables’ measurements. Pest abundance, insecticide usage and yield measurements were collected for 15 years and across all 34 farms, while predator abundance measurements were collected for 9 years at 16 of the 34 farms (4-6 plant-diversified farms and 8-10 mono-rice farms were selected to monitor the predators from 2007 to 2015). This would induce difficulty analyzing the relationship between the predator abundance variable and the other variables inside the structural equation model. The common-location data is another possible data set for path analysis, but potentially have insufficient sample size for fitting structural equation models. While the causal relationships among the variables considered in this study can be further explored through the use of more advanced modeling techniques, but the current data may prohibit us from drawing sound conclusions from such analyses. The path diagram has been added to the revised Supplementary Information (Figure 11—figure supplement 1; Discussion, third paragraph).

Reviewer #2:

[…] My main concerns are related to (1) the selection of the farms, (2) the set-up of the diversified farms, and (3) the insecticide use.

We are grateful for the positive comments, and we have revised our paper in response to the concerns addressed. Two statisticians have been added to our team to perform the statistical model analysis, and we have paid attention to the above concerns proposed by the reviewers in the revised text.

1) It is not clear to me how the farms were selected for this study. Some more information on the selection procedure would be highly appreciated. The sampling design is very unbalanced (28 conventional vs. 6 diversified farms) and not well distributed across Shanghai (the 6 diversified farms were all located on Chongming Island whereas none of the 28 conventional farms was located in this suburb). I guess there must be a good reason for this, which would clarify this slightly awkward sampling design.

As stated above, we have rewritten the text to make all this as clear as possible (Introduction, seventh and last paragraphs).

2) It is unclear to me, how the 6 diversified farms were selected, in particular if these farms were set-up as such through an initiative by those farmers before the observations started or if they were set up specifically for this study through initiative of the researchers. In the latter case, the selection would be highly questionable and could potentially confound most of the results.

Again, we address thus fully in the revised text (Introduction, seventh and last paragraphs).

3) The results of the insecticide use (in particular the number of sprays) seem awkward, given that farmers "applied pesticides according to pest forecast information offered by the Plant Protection Station". I guess this forecast is the same for all farmers and would therefore suggest the same application regime for all farms. I therefore wonder to what extent the differences in pesticide use are simply caused by the contrasting interests of the different farmer communities. A proper pesticide use study would rather apply similar amounts of pesticide and assess their impact on the pest or reduce the pest to similar amounts and quantify pesticide use. However, in this study it seems that both pesticide use and pest abundance varied among farming types, resulting in difficulties to infer sound conclusions.

The use of insecticides in 34 community farms is now clearly described in the text. In this study, pesticide application was not an independent variable, but it is still a very useful variable because pesticides were applied according to the same protocol.

Reviewer #3:

The study analyses a long-term dataset of functional arthropod groups of herbivores and predators (herbivores: three different species of specialist herbivores on rice crop; predators: three generalist predators groups, namely ladybird beetles, lacewings and spiders), replicated over multiple farms and several years. In addition, they have data on insecticide use and yield. The core idea is to test whether a diversified farming system leads to an increased top-down control compared to a monoculture situation, followed by a reduced insecticide use and increased yield. They analysed a total of 6 diversified farms and compared it to 28 control farms. One farm was approximately 2 ha in size. The diversified farm consisted of 9 rice fields surrounded by a stripe of soybeans and an adjacent stripe of another crop. Accordingly, the monoculture farms had 9 rice fields which were surrounded by bare ground. I agree with the authors that we have a lack of knowledge on factors driving ecological intensification in urban systems. However, I have a couple of major concerns with the study.

We appreciate the reviewer’s interest in our paper. We have changed the text in response to the comments.

First, the authors claim that the reduced insecticide use on diversified farms was a consequence of the increased top-down control due to increased predator densities. I wonder how they can rule out that it is not the opposite mechanism, namely that there was a reduced insecticide use on the diversified farms and as a consequence there was an increased predator density which then led to increased top-down control? In other words, how can they rule out that the result they find is not simply a consequence of reduced insecticide use but an effect of having multiple crops on a farm (i.e. diversified farm)?

We appreciate these comments, but there are many reasons to think that our inferences are the most reasonable:

1) When pesticide disturbance was excluded, higher plant species diversity in farming systems has hosted more insect predators, which is supported by a previous study (Wan NF, et al. Chinese Journal of Applied Entomology, 2012, 49: 1604–1609 (in Chinese)). Similar results excluding pesticide disturbance, were reported in our other papers: Wan NF, et al. Ecological Engineering, 2016, 90: 427–430; Wan NF, et al. Ecological Engineering, 2014, 64: 62–65; Wan NF, et al. Ecological Indicators, 2018, in press.

2) In this study, the farmers or workers applied pesticides according to pest abundance that reached the Economic Injury Level (EIL), as we explain in the revised manuscript (subsection “Monitoring sites”, second paragraph). Pesticide application is a response variable, not a true independent variable. Therefore it is extremely unlikely that pesticide application is driving the other results.

A second major concern is the statistical approach used. First, the data are clearly time-series with nicely oscillating densities indicating lag-effects (Figure 2). In addition to lag-effects, the models should include more variables. For example, for explaining predator densities it would be important to incorporate the prey densities in the model.

Thank you for a good question. As stated above, we have redone the analyses with the help of two statisticians. We agree that the farm type effect can be better elucidated by including more predictor variables into the model and by further accounting for possible lag-effects. Unfortunately, the available data may not be sufficient for such comprehensive analyses on the farm-type effect. For instance, the differences in location and time points where the predator abundances and other variables were measured preclude us from obtaining a data set with sufficiently-matched measurements suitable for analyzing the confounding effect of predator abundances on the other variables: pest abundances, insecticide usages, and grain yield were measured across 34 sites over 15 years, yet predator abundance were measured at 16 of the 34 sites across 9 years. Other questions along similar line of thoughts, such as the effect of insecticide usage on the other variables, were ruled out based on arguments mentioned in the response of the reviewer’s previous question. More complex associations among all the variables studied can be analyzed through alternative analytical approaches, and such analyses may be pursued in the future when suitable data can be obtained.

Second, the model (no matter whether it is a time-series analysis or an ANOVA) needs to incorporate the fact that repeated measures were obtained from the same location. Also, to analyse long-term data with multiple t-tests (for each year one test) is not appropriate.

Again, the new statistical analyses respond to these points.

Third, the description of the statistical methods is confusing and incomplete.

See above.

A third major concern is that the main conclusion of higher predator control is not inferred from an experiment (for example, by exposing a standardized number of prey to predators) but based on a correlation.

The concern of the reviewer is correct. This study did not expose a standardized number of prey to predators, but our former study has testified the main conclusion of higher predator in this study (Wan NF, et al. Chinese Journal of Applied Entomology, 2012, 49: 1604–1609). In this literature, when pesticide disturbance was excluded, higher plant species diversity in farming systems has hosted more insect predators.

Finally, the Introduction should be more put into a theoretical context, i.e. why and how do diversified farming systems drive biodiversity and ecosystem functioning. Further, the quality of the literature cited should be improved and all literature cited should be included in the reference list, e.g. Tscharntke et al., 2005.

In response to the helpful comments and those of reviewer 1. The reference (Tscharntke et al., 2005) is very important and we cited it in lots of places. Also, to improve the quality of the literature cited, we have deleted and added some references.

[Editors' note: the author responses to the re-review follow.]

Reviewing Editor (Reviewer #1):

[…] I have no further major comments. It would be good if co-author Jacob Weiner could have a very careful look at the manuscript before the revised version is uploaded again.

Thanks for the encouragements. From the start, Jacob Weiner has been contributing to the manuscript and had a very close look at it before submission.

Reviewer #3:

General comments:[…] I concur with the reviews on the first version of the manuscript – the details of the study, in particular the experimental design and the statistical analysis were unclear. I did not assess the statistical analysis in detail.

We now provide additional information on the statistical analyses in the revision.

Main points:1) I find the title of the paper misleading. The manuscript does not analyse if increasing plant species richness increases biocontrol services etc., because the authors did not analyse diversity in the surroundings of the rice fields. Basically, they show that going away from the monoculture can be beneficial. Thus, the title should be "diversifying agriculture…" or "planting border crops enhances…"

Thanks for the useful suggestion. We changed the title to “Increasing plant diversity with border crops reduces insecticide use and increases crop yield in urban agriculture”.

2) Nowhere in the Results are a test statistic or the number of replicates are given.

We now give this information.

3) Materials and methods: it is unclear how information on insecticide use was obtained.

In the 15-year monitoring data, the farmers in the 34 community farms uploaded the pest information through the server into the Shanghai Pest Monitoring System which was accessible to the monitoring locations, 9 districts of Agricultural Technology Extension and Service Centers and SATESC (subsection “Monitoring sites”, fourth paragraph). In the common-location experiments, we kept a written record of the insecticide input for each plot, and this was used to obtain the information on insecticide use (subsection “Common-location experiments”, third paragraph).

4) Materials and methods: unfortunately, I find that still many details are unclear in the revision, see below.

We have made the Materials and methods section much more detailed.

5) Results: the effect on yield is only marginally significant and should not be overstated. Also, the authors provide no mechanism why yield was higher – do the insecticides not work?

In this revision, we have mentioned that “marginally higher grain yield”. Besides, we have provided the mechanism why yield was higher in the Discussion: “In all likelihood, the reduced pest abundance contributed to rice growth, development and reproduction, and thus increased rice yield”. “Thus, the most likely explanation is that the increase in the abundance of invertebrate predators spread from border and neighboring crop areas onto the rice fields, increasing predation on pest populations”.

Materials and methods:Subsection “Monitoring sites”: the entire section should be restructured, the design of the diversified fields is described twice (first and third paragraphs), the same is true for other information. Start with the region, then describe the farms, then how rice is grown and then the different treatments.

Thanks for the suggestion. We have restructured this section (subsection “Monitoring sites”).

Subsection “Monitoring sites”, first paragraph: please provide a drawing of a diversified field, and a photograph, how large is the area where other crops are grown.

We have added a drawing of a diversified field, and a photograph (see Figure 13—figure supplement 1).

Subsection “Monitoring sites”, second paragraph: the pest control guidance by SATESC is not known to the reader – what are the insecticides used, what active ingredients etc.?

The names and active ingredients of the insecticide used were listed in this revision (see Figure 1—source data 2).

Subsection “Monitoring sites”, last paragraph: what is a "Z" subplot? Again, use a drawing.

The drawing related to "Z" subplot has been presented in this revision (see Figure 12—figure supplement 1).

Subsection “Common location experiments”, first paragraph: design unclear: what is the replicate – a block, i.e. three pairs of?

For each site, the experiment was a randomized block design with three replicate blocks. Each block contained two plots – one for plant-diversified farming and the other for mono-rice farming (50-55 m × 30-35 m plot size in Xinchang, and 55-60 m × 35-40 m in Sanxing) (subsection “Common-location experiments”, first paragraph).

Subsection “Common location experiments”, fourth paragraph etc.: Please provide more detail of the pest assessments: I assume the light kills insects attracted to the light? Were they collected in alcohol? Who identified the species?

Insect trapping lamps were installed in the periphery of 9 paddy plots of each community farm to measure the abundance of the main herbivore pests. The lamps utilized light to attract the pests to fall into the cloth bags or cylindrical iron buckets of the lamp, and the workers in each farm identified the species and counted the attracted pests in the bags or buckets each day (subsection “Monitoring and sampling methods”, first paragraph). The sets of trapping insect herbivores are as shown in Author response image 1.

Why was the interval for the planthopper different if the same lamp in the same field is used?

The life histories and dates of occurrence on rice fields of pink rice borers and planthoppers are different. Thus, the monitoring intervals are different. From 2001 to 2015, we monitored the number of trapped pink rice borers and rice brown planthoppers from 10 April to 30 September and from 11 May to 30 September, respectively, as this was when they occurred in the rice fields (subsection “Monitoring and sampling methods”, first paragraph).

When you say each rice field, do you mean one in each of 9 plots per farm, or just one?

There were some errors in the previous submission. Insect trapping lamps were installed in the periphery of 9 paddy plots of each community farm to measure the abundance of the main herbivore pests (pink rice borer, rice brown planthopper, etc.) (subsection “Monitoring and sampling methods”, first paragraph).

Subsection “Monitoring and sampling methods”, last paragraph: did you only count predators on plants?

We only counted predators of the pests on the plants.

Subsection “Common location experiments”, fourth paragraph: what is the correlation between number of pests and number of predators? I assume it is positive? Did you also count lacewing larvae?

We analyzed the correlation between number of pests and number of predators, and found that the relationship to be significantly positive (see Author response image 2). However, we think that this figure has no role in this study, so we did not present it in the revision. Additionally, we also counted lacewing larvae (subsection “Common-location experiments”, last paragraph).

**Author response image 2. respfig2:** The correlation between number of insect pests and number of predators in rice field plots. Here insect pests included rice planthoppers, rice stem borers, and rice leaf rollers; predators were involved in ladybird beetles, lacewings and spiders.

I also do not understand how the farmers decided when to spray – you state that there are economic thresholds, so there must be data on how often the threshold was reached? Were the pest data used for the threshold analysis the same that you used?

Economic Injury Levels of pink rice borers, rice brown planthoppers and rice leaf rollers are listed in Figure 2—source data 4.

Figures:Figure 1: I thought the 28 farms were mono-rice and only 6 diversified, not the other way round?

We have revised the caption of Figure 1.

Figure 2: were pest insects sampled all year round, or only from June onwards? This is not stated in the manuscript. In the first paragraph of the subsection “Monitoring and sampling methods” it states sampling was done May to September (April to September).

We have added the corresponding information in Figure 2.

Figure 5: 9-14 kg insecticide per hectare is a lot – what is this number referring to, the liquid that is brought to the field? Details on the insecticides should be given and the amount plotted in active ingredient rather than some other unit.

The insecticide per hectare that is brought to the field is the liquid or powder (amount of commercial insecticide). We think that commercial and active ingredient insecticide uses are both important indicators to analyze the ecological intensification. Thus, we provided another Figure 5C.